# FIBER: A Differentially Private Optimizer with Filter-Aware Innovation Bias Correction

**Duc Dm** [1]  **Thao Do** [1]  **Minh Son Hoang** [1]  **Tran Le Duc Anh** [2]  **Daeyoung Kim** [† 1]  **Huy L. Nguyen** [† 3]

## Abstract

Differentially private (DP) training protects individual examples by adding noise to gradients, but the injected noise interacts nontrivially with adaptive optimizers. Recent DP methods temporally filter privatized gradients to reduce variance; however, filtering also changes the DP noise statistics seen by AdamW's second-moment accumulator. As a result, bias corrections derived for unfiltered DP noise (e.g., subtracting $\sigma_w^2$) can become miscalibrated when filtering is present. We propose FIBER, a DP optimizer designed for temporally filtered privatized gradients. FIBER (i) performs denoising in innovation space by filtering the residual stream and integrating it to form the filtered gradient estimate, (ii) decouples the two-point observation geometry from the innovation gain to enable independent tuning, and (iii) introduces a filter-aware second-moment calibration that subtracts the attenuated DP noise contribution $A(\omega)\sigma_w^2$, where $A(\omega)$ is derived in closed form for the innovation filter and can be computed for general stable linear filters. Across vision and language benchmarks, FIBER consistently demonstrates substantial improvements in the performance of DP optimizers, surpassing state-of-the-art results under equivalent privacy constraints on multiple tasks.

## 1. Introduction

Differential privacy (Dwork & Roth, 2014) offers strong protection for individual data in machine learning, but it often reduces model utility, especially during long train-ing, with high-dimensional models, or in fine-tuning where optimization is sensitive (Abadi et al., 2016; Jayaraman & Evans, 2019; De et al., 2022).

A central challenge is that DP noise does not remain isolated in the gradient it is added to. In modern adaptive optimizers, privatized gradients are aggregated into momentum and second-moment statistics, so DP perturbations become stateful through the optimizer state. For Adam (Kingma & Ba, 2015), the squared-gradient accumulator $v_t = \beta_2 v_{t-1} + (1 - \beta_2)g_t^2$ is nonlinear and converts zero-mean DP noise into persistent positive bias. This preconditioner inflation shrinks effective step sizes and degrades adaptivity over time. Tang et al. (2024) show that DP-Adam can collapse toward DP-SGD behavior unless this bias is explicitly handled.

Recent work applies temporal filtering to privatized gradients to reduce DP noise impact. Methods range from low-pass filters (Zhang et al., 2024a;b) to correlated-noise mechanisms (Kairouz et al., 2021; Choquette-Choo et al., 2024; Koloskova et al., 2023), showing clear benefits for gradient estimation. However, filtering introduces a previously unmodeled failure mode when combined with adaptive optimizers. The core issue is that filtering attenuates DP noise variance before it reaches the second-moment accumulator. Temporal denoising changes the DP noise statistics seen by adaptive optimizers. Let $g_t = \bar{g}_t + w_t$ with $w_t \sim \mathcal{N}(0, \sigma_w^2 I_d)$, $\bar{g}_t$ denote the clipped minibatch gradient estimate before DP noise is added and $g_t$ be the privatized gradient used by the learning algorithm, and let $\tilde{g}_t$ denote a filtered version used to update AdamW moments. For stable linear filters, the DP noise component is attenuated. In steady state, $\mathrm{Var}(\tilde{g}_{t,i}|\bar{g}) = A\,\sigma_w^2$ for some $A \in (0, 1]$. Consequently, the bias-corrected second moment satisfies the approximation $\mathbb{E}[\hat{v}_{t,i}] \approx \mathbb{E}[\tilde{g}_{t,i}^2] = \mathbb{E}[s_{t,i}^2] + A\,\sigma_w^2$, where $s_t$ is the filtered signal component. This has two implications: (i) without correction, $\hat{v}_t$ retains a positive DP noise term that can shrink adaptive steps, and (ii) DP-Adam bias corrections derived for unfiltered noise (subtracting $\sigma_w^2$) become miscalibrated when filtering is present. We therefore propose a filter-aware correction that subtracts the attenuated DP noise contribution $A\,\sigma_w^2$, yielding a better-calibrated preconditioner under filtered DP noise. Existing filtering methods (Zhang et al., 2024a;b; Koloskova et al., 2023)

†Equal supervision. [1] School of Computing, Korea Advanced Institute of Science and Technology, Daejeon, Republic of Korea [2] Center for Environmental Intelligence (CEI), VinUniversity, Hanoi, Vietnam [3] Khoury College of Computer Sciences, Northeastern University, USA . Correspondence to: Duc Dm <ducdm200158@kaist.ac.kr>.

improve gradient estimation but do not model or correct how filtering alters DP noise statistics within AdamW's internal state, leaving a critical gap. We introduce FIBER (**F**ilter-aware **I**nnovation **B**ias-corrected optimiz**ER**), a differentially private adaptive optimizer tailored to temporally filtered privatized gradients. Our main contributions are:

- **Principled innovation-space filtering.** We propose a denoising approach in residual space, integrating the residual stream with a lightweight, theoretically grounded second-order recursion. This method enables tracking of nonstationary dynamics under DP noise.

- **Decoupled hyperparameter control.** We provide the first explicit separation of observation geometry $(\kappa, \gamma)$ from innovation gain $\omega$, enabling independent tuning of estimator geometry and temporal smoothing for practical and effective hyperparameter optimization.

- **First filter-aware DP-AdamW calibration.** We rigorously analyze how temporal filtering affects the statistics of DP noise and derive the precise attenuation factor. To our knowledge, this is the first explicit analysis and practical calibration of AdamW's second-moment statistics under filtered DP noise, including an attenuation-aware correction for stable linear filters.

- **Extensive empirical validation.** FIBER consistently outperforms DP-Adam(W) and temporal-filtering baselines across a wide range of vision and language benchmarks, achieving state-of-the-art performance under tight privacy budgets and long training horizons, while remaining competitive at higher $\varepsilon$.

**Position relative to prior filtering methods.** Although FIBER employs the same two-point observation interface as DiSK, it differs in several important respects. First, FIBER filters the innovation or residual stream rather than the gradient state. Second, it decouples the two-point geometry parameters $(\kappa, \gamma)$ from the denoising gain $\omega$. Third, it calibrates the second moment in AdamW using the post-filter differential privacy noise variance $A(\omega)\sigma_w^2$. Therefore, the primary contribution of FIBER lies not solely in the use of two-point gradients, but in the integration of innovation-space denoising with filter-aware adaptive preconditioning.

## 2. Related Work

### 2.1. Differentially Private Optimization Methods

Most practical DP training algorithms use per-example gradient clipping and add Gaussian noise, resulting in DP-SGD and adaptive variants such as DP-Adam and DP-AdamW (Abadi et al., 2016; Yu et al., 2024; Gilani et al., 2025). While effective in smaller regimes and some fine-tuning

tasks, these methods often suffer in long-horizon training and large-scale models (Jayaraman & Evans, 2019; De et al., 2022). Another line of work improves robustness to DP perturbations through mechanism- and training-level choices, including adaptive clipping and automated tuning (Bu et al., 2023a; Xia et al., 2023), as well as architectural and optimization design choices that reduce sensitivity to DP noise (Yu et al., 2024; Bu et al., 2023b; Mehta et al., 2023). For adaptive optimizers in particular, recent work analyzes how DP noise biases the internal second-moment accumulator and can collapse adaptivity over time, motivating explicit bias-correction strategies (Tang et al., 2024). The present work is aligned with this perspective but focuses on a setting that is increasingly prevalent in practice: filtered privatized gradients. In this regime, the statistics entering AdamW are altered by the filter, and correcting the second moment requires accounting for the filter-induced attenuation, not only the raw DP noise level.

### 2.2. Temporal Structure: Filtering and Correlation

Recent research leverages the temporal structure of DP noise to improve optimization. Correlated-noise mechanisms show that coupling noise across iterations can improve privacy-utility trade-offs in certain regimes (Kairouz et al., 2021; Choquette-Choo et al., 2024; Koloskova et al., 2023). In parallel, signal-processing-inspired methods apply temporal low-pass filtering to privatized gradients to suppress high-frequency DP noise while preserving learning signal (Zhang et al., 2024b). Several approaches also address clipping-induced distortion (e.g., via error-feedback-like corrections) to recover unbiased optimization behavior under clipping constraints (Zhang et al., 2024c). Some view DP training as filtering: privatized gradients are noisy measurements of an underlying time-varying learning signal, and temporal structure is exploited to suppress injected noise while preserving signal. Kalman-filter optimizers are impractical for deep learning due to costly covariance tracking and stepwise matrix operations (Vuckovic, 2018). Recent work favors constant-gain, lightweight state-space variants. DiSK, for example, uses a two-point measure, computing gradients at both current $\theta_t$ and lookahead parameters $\theta_t + \gamma d_{t-1}$ and exponential smoothing of a latent gradient state-a first-order model with an EMA filter-providing empirical and theoretical guarantees (Zhang et al., 2024a).

Despite recent advances, a crucial challenge persists: temporal filtering can miscalibrate adaptive preconditioners because the noise statistics observed by AdamW's moments fluctuate with privacy budgets and training dynamics. Consequently, naive smoothing or variance subtraction may lead to unstable updates. This motivates protocols to test robustness across regimes and methods that go beyond gradient-state heuristics by aligning optimizer moments with the post-filter noise model.

# 3. Method

## 3.1. Problem Setup

Given a dataset $\mathcal{D} = \{\xi_i\}_{i=1}^N$, we minimize empirical risk:

$$\min_{\theta \in \mathbb{R}^d} F(\theta) \triangleq \frac{1}{N} \sum_{i=1}^N f(\theta; \xi_i), \tag{1}$$

where $f(\theta; \xi)$ is the per-example loss and $\theta$ are parameters. At iteration $t$, sample minibatch $B_t \subset \mathcal{D}$ of size $B$.

## 3.2. Differentially Private Optimization

We briefly review $(\varepsilon, \delta)$-differential privacy(Dwork & Roth, 2014), the Gaussian mechanism(Dwork & Roth, 2014; Wang et al., 2019), and DP-SGD-the standard privatization procedure used in DP optimization. Background details are deferred to Appendix A.

**DP two-point gradient observation.** At iteration $t$, a privatized gradient observation is formed using a two-point per-example construction, followed by per-example clipping and Gaussian noise, with $\text{clip}(u, C) = u \cdot \min\{1, C/\|u\|_2\}$:

$$u_t(\xi) \triangleq a \nabla f(\theta_t + \gamma d_{t-1}; \xi) + (1-a) \nabla f(\theta_t; \xi), \tag{2}$$

$$g_t \triangleq \frac{1}{B} \sum_{\xi \in B_t} \text{clip}(u_t(\xi), C) + w_t. \tag{3}$$

Here $a \triangleq \frac{1-\kappa}{\kappa\gamma}$, $d_{t-1} \triangleq \theta_t - \theta_{t-1}$, we set $d_{-1} = 0$ and $w_t \sim \mathcal{N}(0, \sigma_w^2 I_d)$ with $\sigma_w^2$ the per-coordinate variance at the averaged gradient. DiSK (Zhang et al., 2024a) denoises the privatized gradient $g_t$ with an EMA filter:

$$\tilde{g}_t \triangleq (1-\kappa)\tilde{g}_{t-1} + \kappa g_t, \tag{4}$$

This filtered gradient is used by the optimizer. To avoid negative weights (which can amplify norms and disrupt DP clipping), we require $0 \le a \le 1$; for $\kappa \in (0, 1)$ and $\gamma > 0$, this is equivalent to $\gamma \ge (1-\kappa)/\kappa$ (see Appendix E.5).

## 3.3. FIBER: Innovation-Filtered DP-AdamW with Filter-Aware Bias Correction

FIBER is structured around three primary components: (i) innovation-space filtering, (ii) decoupling the two-point estimator from the denoiser, and (iii) filter-aware bias correction for AdamW under filtered DP noise. FIBER preserves DP-SGD's privacy guarantees. Privacy preservation follows from post-processing: all FIBER computations are deterministic functions of the privatized gradients $\{g_t\}$, so by the composition theorem of DP (Dwork & Roth, 2014), if $\{g_t\}$ is $(\varepsilon, \delta)$-DP, then FIBER's output is also $(\varepsilon, \delta)$-DP.

**Component 1: Innovation-space filtering.** Rather than directly smoothing the gradient state (4), FIBER filters the residual process

$$\nu_t \triangleq g_t - \tilde{g}_{t-1}, \tag{5}$$

This approach maintains a smoothed residual state $r_t$ and integrates it to obtain the denoised estimate $\tilde{g}_t$:

$$r_t \triangleq (1-\omega) r_{t-1} + \omega \nu_t, \tag{6}$$

$$\tilde{g}_t \triangleq \tilde{g}_{t-1} + r_t. \tag{7}$$

Gradient-state EMA smoothing applies a first-order low-pass filter directly to $g_t$, which trades noise suppression for lag when the underlying gradient drifts. In contrast, the $\alpha$–$\beta$ or residual view smooths only the residual $\nu_t$ and integrates it into $\tilde{g}_t$. This yields a second-order recursion that can track persistent drift in the latent gradient. This distinction corresponds to different latent models (random-walk vs. constant-velocity) and is made formal in Section 4.1.

**Component 2: Decoupling $(\kappa, \gamma)$ from $\omega$.** The two-point parameters $(\kappa, \gamma)$ appear only in the observation construction (2) (as determined by $a = \frac{1-\kappa}{\kappa\gamma}$), while the denoising gain $\omega$ appears only in the residual recursion (6)–(7). This separation enables independent control of extrapolation geometry ($(\kappa, \gamma)$) and temporal denoising strength ($\omega$), instead of coupling both behaviors to a single gain parameter.

**Component 3: Filter-aware bias correction for AdamW.** AdamW moments use the denoised $\tilde{g}_t$:

$$m_t \triangleq \beta_1 m_{t-1} + (1-\beta_1)\tilde{g}_t, \tag{8}$$

$$v_t \triangleq \beta_2 v_{t-1} + (1-\beta_2)(\tilde{g}_t \odot \tilde{g}_t), \tag{9}$$

with bias corrections $\hat{m}_t = m_t/(1-\beta_1^{t+1})$ and $\hat{v}_t = v_t/(1-\beta_2^{t+1})$. With decoupled weight decay $\lambda$, the AdamW update is

$$\theta_{t+1} = (1-\eta\lambda)\theta_t - \eta \hat{m}_t \oslash (\sqrt{\bar{v}_t} + \epsilon), \tag{10}$$

where $\bar{v}_t$ is a corrected second moment. Since the DP perturbation $w_t$ in (3) is filtered before it enters the second-moment recursion, the resulting variance in the filtered gradient $\tilde{g}_t$ is reduced by a factor $A(\omega)$ (Appendix C). We therefore subtract the expected filtered contribution from the bias-corrected second moment:

$$\sigma_{\text{filt}}^2 \triangleq A(\omega) \sigma_w^2, \qquad A(\omega) \triangleq \frac{2-\omega}{4-3\omega}. \tag{11}$$

We then define the corrected accumulator

$$\bar{v}_t \triangleq \max(\hat{v}_t - \sigma_{\text{filt}}^2, \epsilon_v), \tag{12}$$

where $\epsilon_v \ge 0$ is a small floor for numerical stability. Since (11) uses steady-state variance, early iterations may be conservative; the floor prevents negative and over-corrected values. If signal and DP noise are approximately uncorrelated, this adjustment removes filtered variance - though cross terms may persist in closed-loop training (Appendix E.4).

**Algorithm 1** Simplified FIBER (full pseudocode in Appendix D)

1: **Input:** $\theta_0$, steps $T$.
2: **DP params:** batch $B$, clip $C$, noise $\sigma_{DP}$.
3: **AdamW params:** step $\eta$, $(\beta_1, \beta_2)$, $\epsilon$, weight decay $\lambda$.
4: **FIBER params:** $(\kappa, \gamma, \omega, \epsilon_v)$.
5: **State init:** $\tilde{g}_t \leftarrow 0$, $r \leftarrow 0$, $m \leftarrow 0$, $v \leftarrow 0$.
6: **Constants:** $\sigma_w^2 \leftarrow (\sigma_{DP} C/B)^2$; $A(\omega)$ as in Eq. (11).
7: **for** $t = 0, \ldots, T-1$ **do**
8:     **(1) DP observation**
9:       Form $g_t$ via two-point observation + clipping + Gaussian noise (Eqs. (2)–(3)).
10:     **(2) Innovation (residual) filtering**
11:       $\nu_t \leftarrow g_t - \tilde{g}_{t-1}$
12:       $r_t \leftarrow (1-\omega)r_{t-1} + \omega\nu_t$
13:       $\tilde{g}_t \leftarrow \tilde{g}_{t-1} + r_t$
14:     **(3) AdamW moments**
15:       $m_t \leftarrow \beta_1 m_{t-1} + (1-\beta_1)\tilde{g}_t$
16:       $v_t \leftarrow \beta_2 v_{t-1} + (1-\beta_2)(\tilde{g}_t \odot \tilde{g}_t)$
17:     **(4) Bias + filter-aware second-moment correction**
18:       $\hat{m}_t \leftarrow m_t/(1-\beta_1^{t+1})$
19:       $\hat{v}_t \leftarrow v_t/(1-\beta_2^{t+1})$
20:       $\bar{v}_t \leftarrow \max(\hat{v}_t - A(\omega)\sigma_w^2, \epsilon_v)$
21:     **(5) Parameter update**
22:       $\theta_{t+1} \leftarrow (1-\eta\lambda)\theta_t - \eta\hat{m}_t \oslash (\sqrt{\bar{v}_t} + \epsilon)$
23:       $d \leftarrow \theta_{t+1} - \theta_t$
24: **end for**

**Pseudocode.** Algorithm 1 summarizes FIBER. The two-point per-example observation $u_t(\xi)$ is defined in Equation (2); clipping and privatization follow Equation (3).

## 4. Theoretical Analysis

### 4.1. Residual Gradients and Innovation Filtering

Our first contribution is a residual-gradient dynamics viewpoint, which yields Equations (5)–(7) as a simplified constant-gain Kalman filter; we further validate its drift-tracking behavior via diagnostics in Appendix E.3.

**Proposition 4.1.** *Consider a constant-velocity model for the latent (noise-free) gradient in which the gradient has a slowly varying drift component. In steady state, the Kalman filter yields a constant-gain $\alpha$–$\beta$ recursion in residual form with gains $(\alpha, \beta)$. To reduce hyperparameter tuning and avoid covariance tracking,* FIBER *uses a tied-gain approximation $\alpha = \beta = \omega$, recovering (5)–(7).*

FIBER filters innovations and integrates them to denoise gradients. Setting $\alpha = \beta = \omega$ yields a single bias-variance knob; this is a tied-gain, constant-gain surrogate motivated by rapid gain convergence and avoiding covariance tracking, though it generally does not match the optimal steady-state Kalman gains. See Appendix B.1 for details.

### 4.2. Decoupling Estimation and Smoothing

Our second contribution is to decouple the two-point estimator parameters $(\kappa, \gamma)$ (which determine how per-example gradients are formed before clipping/noise) from the residual smoother gain $\omega$ (which controls temporal denoising after privatization).

*Remark 4.2.* In single-gain two-point formulations that also use gradient-state smoothing, the same parameter $\kappa$ controls both (i) temporal smoothing ($\tilde{g}_t = (1-\kappa)\tilde{g}_{t-1} + \kappa g_t$) and (ii) the two-point mixing weight $a(\kappa, \gamma)$, so changing $\kappa$ necessarily changes both effects. In FIBER, $(\kappa, \gamma)$ appear only in the construction of $u_t(\xi)$, while $\omega$ appears only in the residual recursion (5)-(7). This separation enables independent tuning of extrapolation geometry and temporal denoising, which we validate empirically in Section 5.3.

### 4.3. Filter-Aware Bias Correction for AdamW

Our third contribution corrects AdamW's second-moment estimate under filtered DP noise. Recall that the privatized gradient can be written as $g_t = \bar{g}_t + w_t$, where $w_t$ is the Gaussian DP noise. FIBER applies the residual filter (5)-(7) to $g_t$, which turns i.i.d. noise into a colored sequence but attenuates its marginal variance by a closed-form factor.

**Proposition 4.3.** *Let $\{w_t\}$ be i.i.d. $\mathcal{N}(0, \sigma_w^2)$, and an LTI filter with impulse response $\{h_j\}$ outputs*

$$\tilde{w}_t = \sum_{j=0}^{\infty} h_j w_{t-j}. \qquad (13)$$

*If the filter is $\ell_2$-stable ($\sum_j h_j^2 < \infty$), then at stationarity*

$$\mathrm{Var}(\tilde{w}_t) = \sigma_w^2 \sum_{j=0}^{\infty} h_j^2 \triangleq A\sigma_w^2, \qquad (14)$$

*where $A$ is the squared $\ell_2$ gain of the filter. For vector DP noise $w_t \sim \mathcal{N}(0, \sigma_w^2 I_d)$ filtered coordinate-wise,*

$$\mathbb{E}\|\tilde{w}_t\|_2^2 = d A \sigma_w^2. \qquad (15)$$

*Proof.* Using (13) and independence of $\{w_t\}$, all cross terms vanish, so $\mathbb{E}[\tilde{w}_t^2] = \sum_{j\geq 0} h_j^2 \mathbb{E}[w_{t-j}^2] = \sigma_w^2 \sum_{j\geq 0} h_j^2$. The vector case follows by summing coordinate-wise variances. $\square$

**Corollary 4.4.** *Consider a stable linear state-space system driven by scalar $w_t \sim \mathcal{N}(0, \sigma_w^2)$:*

$$z_t = Mz_{t-1} + Gw_t, \qquad \tilde{w}_t = Hz_t, \qquad (16)$$

*with $\rho(M) < 1$. Let $\Sigma$ be the stationary state covariance solving the Lyapunov equation*

$$\Sigma = M\Sigma M^\top + \sigma_w^2 GG^\top. \qquad (17)$$

*Then $\mathrm{Var}(\tilde{w}_t) = H\Sigma H^\top \triangleq A\sigma_w^2$.*

Applying Corollary 4.4 to the state-space realization of the residual filter yields the closed-form attenuation.

**Corollary 4.5.** *For the innovation recursion in Equations (5)-(7) with gain $\omega \in (0, 1]$ driven by i.i.d. DP noise,*

$$A(\omega) = \frac{2 - \omega}{4 - 3\omega} \in \left[\frac{1}{2}, 1\right]. \tag{18}$$

**Proposition 4.6.** *Assume the input to the innovation filter is pure i.i.d. Gaussian noise, i.e., $g_t = w_t$ with $w_t \sim \mathcal{N}(0, \sigma_w^2 I_d)$ independent across $t$. Then for any $\omega \in (0, 1]$, the innovation filter (5)-(7) is stable and, in steady state, each coordinate satisfies*

$$\lim_{t \to \infty} \mathrm{Var}(\tilde{g}_{t,i}) = A(\omega)\, \sigma_w^2 \tag{19}$$

*Remark* 4.7. Any stable linear temporal filter scales the marginal variance of i.i.d. DP noise by its squared $\ell_2$ gain $A = \sum_{j \geq 0} h_j^2$ (or via the Lyapunov equation). Thus, AdamW's second-moment estimate is corrected by subtracting $A\sigma_w^2$ per coordinate, with a floor $\epsilon_v$. Appendix C gives full derivations, including finite-time bounds and second-moment decomposition for innovation filters, and discusses possible cross terms under closed-loop training.

## 5. Experiments

We evaluate FIBER on a diverse set of computer vision (CV) and natural language processing (NLP) tasks under differential privacy. We test if innovation-space denoising and filter-aware second-moment correction improve convergence and performance in practical DP settings. Implementation details are in Appendix E, hyperparameter sensitivity in Appendix E.5, and multi-seed results in Appendix F.8.

### 5.1. Experimental Setup

**Privacy accounting and hyperparameters.** All results use subsampled Gaussian mechanism with per-example $\ell_2$ clipping at norm $C$. The noise multiplier is computed using a Rényi DP (RDP) (Wang et al., 2019; Bu et al., 2023b) accountant with fixed batch size sampling without replacement. We set $\delta = 1/N^{1.1}$ for privacy. The default $\epsilon_v = 10^{-8}$ is justified in Appendix E.6.

**Datasets and tasks.** For vision, we train on MNIST (LeCun et al., 1998), CIFAR-10/100 (Krizhevsky, 2009), and ImageNet-1k (Deng et al., 2009). For NLP, we fine-tune on GLUE (Wang et al., 2018) and evaluate text generation on E2E (Novikova et al., 2017).

**Models.** We use CNN5 for MNIST/CIFAR-10, WRN for CIFAR-100, and ViT-small (Dosovitskiy et al., 2020) for ImageNet-1k. For NLP, we fine-tune RoBERTa (Liu et al., 2019) on GLUE and GPT-2-small (Radford et al., 2019) on

E2E. More results on multiseed are reported in Appendix E Models are trained from scratch or fine-tuned from HuggingFace checkpoints (Wolf et al., 2020).

**Hyperparameter tuning.** For each task and privacy budget, we independently tune all hyperparameters for each optimizer using 100 grid search trials per optimizer. Base hyperparameters (learning rate, epochs, batch size) are not shared between methods. Baseline-specific parameters (see Table 10) are tuned with the same budget as FIBER. For FIBER, we stage the search: first tuning $(\kappa, \gamma)$, then $\omega$ using the best $(\kappa, \gamma)$ pair. Unless noted, $(\kappa, \gamma, \omega) = (0.6, 0.7, 0.9)$.

**Baselines.** We design baselines to span the main mechanisms used to improve DP training, so that each claim is evaluated against an appropriate control. We include unfiltered adaptive DP optimizers - DPAdamW with per-example clipping and Gaussian noise (we use DPAdam for CV tasks); temporal filtering baselines, DiSK (Zhang et al., 2024a) and DOPPLER (Zhang et al., 2024b); a correlated-noise baseline, MF-DP-FTRL (Choquette-Choo et al., 2023), which introduces structured correlations across steps/epochs; and a compositional baseline, DiSK-CORR, which augments DiSK with our filter-aware second-moment correction. For EMA state filtering with i.i.d. noise, the steady-state attenuation is $A_{\mathrm{state}}(\kappa) = \kappa/(2 - \kappa)$; accordingly, we subtract $A_{\mathrm{state}}(\kappa)\sigma_w^2$ from $\hat{v}_t$ (details in Appendix C.4).

### 5.2. Main Results

**CV tasks.** Figure 1 shows test accuracy across privacy budgets for MNIST (CNN5), CIFAR-10 (CNN5), and CIFAR-100 (WRN). Across all datasets, FIBER consistently outperforms DiSK, with the largest gains at low $\varepsilon$, where DP noise dominates. We also compare temporal filtering baselines: DiSK and DiSK-CORR. DiSK-CORR often matches or improves on DiSK, supporting our point that adaptive preconditioning must be calibrated to filtered DP noise. FIBER remains best, indicating benefit from innovation-space denoising beyond second-moment correction. For CIFAR-10, we also include baselines: DOPPLER (low-pass DP-training) and MF-DP-FTRL (correlated-noise baseline). Figure 2 shows representative learning curves at fixed privacy budgets. FIBER converges faster and reaches higher accuracy than DPAdam and DiSK. We also evaluate ViT-small on ImageNet-1k under DP; the learning curve is shown in Figure 2, with FIBER outperforming DiSK and DPAdam, raising accuracy from 39 to 44. Early gains suggest FIBER mitigates DP-induced optimization slowdown. We further evaluate parameter transfer by finetuning a pretrained ViT-small on CIFAR-100. Table 1 presents accuracy results across varying privacy budgets. FIBER demonstrates superior performance compared to both DP-

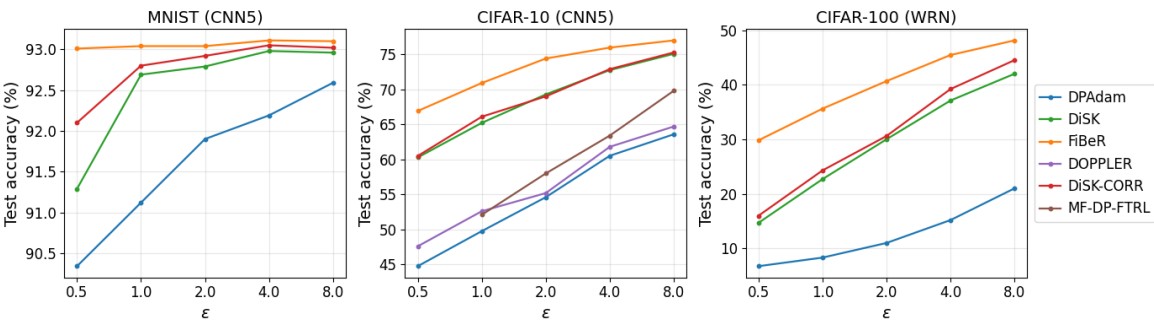

*Figure 1.* **Training from scratch across privacy budgets.** Final test accuracy on MNIST (CNN5), CIFAR-10 (CNN5), and CIFAR-100 (WRN) under different $\varepsilon$. We compare DPAdam, DiSK, DiSK-CORR, and FIBER. On CIFAR-10 (CNN5) we additionally include DOPPLER as a low-pass baseline and MF-DP-FTRL as a correlated-noise baseline.

Adam and DiSK for all values of $\varepsilon$, with the most substantial improvements observed under stricter privacy constraints. Relative to DiSK, FIBER increases accuracy by $+2.0$ to $+5.1$ percentage points as $\varepsilon$ ranges from 1 to 8, with greater improvements at lower privacy budgets. These results indicate that innovation filtering and filter-aware correction remain effective during finetuning, even when pretrained representations reduce optimization difficulty.

*Table 1.* Fine-tuning ViT-small on CIFAR-100 under DP. (**Best** and second best are highlighted.)

| Method | $\varepsilon = 0.5$ | $\varepsilon = 1.0$ | $\varepsilon = 2.0$ | $\varepsilon = 4.0$ | $\varepsilon = 8.0$ |
|---|---|---|---|---|---|
| DP-AdamW | 63.2 | 78.0 | 83.7 | 85.7 | 86.8 |
| DiSK | 83.5 | 85.4 | 86.8 | 87.6 | 88.5 |
| FIBER | **88.6** | **89.4** | **90.0** | **90.4** | **90.5** |

**NLP tasks.** We fine-tune RoBERTa-base on MNLI, QNLI, SST-2, and QQP with DP. FIBER consistently outperforms DPAdamW across all tasks and privacy budgets, especially at $\varepsilon=1$. Its denoising and calibration methods improve language fine-tuning under DP. FIBER is usually competitive with or better than DiSK, demonstrating the value of innovation filtering. DOPPLER results use looser privacy budgets. See Appendix E for more NLP results and settings.

*Table 2.* Task metric of fine-tuning result on the GLUE dataset.

| Task | $\varepsilon$ | Non-DP | DPAdamW | DiSK | DOPPLER | FIBER |
|---|---|---|---|---|---|---|
| MNLI | 6.7 | 87.6 | 83.2 | **84.8** | 83.80 | 84.7 |
| | 1.0 | 87.6 | 80.7 | 82.0 | **83.55** | 83.0 |
| QNLI | 6.7 | 92.8 | 87.5 | 88.9 | 87.76 | **90.6** |
| | 1.0 | 92.8 | 86.0 | 88.7 | 87.63 | **90.5** |
| SST-2 | 6.7 | 94.8 | 91.5 | 92.8 | 91.82 | **94.0** |
| | 1.0 | 94.8 | 91.4 | 91.5 | 91.71 | **93.1** |
| QQP | 6.7 | 91.9 | 85.8 | 89.0 | 86.50 | **89.5** |
| | 1.0 | 91.9 | 84.2 | 86.9 | 85.71 | **88.6** |

**NLG tasks** We evaluate FIBER for DP text generation by fine-tuning GPT-2-small with HuggingFace checkpoints on the E2E dataset. Our task setup, training scripts, and hyperparameter tuning protocol follow those of previous DP

generation studies (e.g., Li et al., 2022) to ensure comparability. We set $(\kappa, \gamma, \omega) = (0.6, 0.7, 0.9)$. Table 3 reports results on E2E for $\varepsilon \in \{3, 8\}$. FIBER consistently improves over DPAdamW and achieves performance close to DiSK, while exhibiting slightly different metric trade-offs.

*Table 3.* GPT-2 fine-tuning on E2E (higher is better).

| Algorithm | BLEU (%) | ROUGE-L (%) | METEOR | NIST | CIDEr |
|---|---|---|---|---|---|
| AdamW ($\varepsilon = \infty$) | 69.46 | 71.36 | 0.461 | 8.780 | 2.422 |
| DPAdamW ($\varepsilon = 3$) | 61.52 | 65.87 | 0.417 | 7.071 | 2.167 |
| DiSK ($\varepsilon = 3$) | **68.35** | **70.23** | 0.456 | 8.636 | 2.399 |
| FIBER ($\varepsilon = 3$) | 67.57 | 69.97 | **0.463** | **8.660** | **2.407** |
| DPAdamW ($\varepsilon = 8$) | 64.99 | 67.34 | 0.425 | 8.387 | 2.192 |
| DiSK ($\varepsilon = 8$) | **68.73** | **70.58** | 0.460 | **8.697** | **2.463** |
| FIBER ($\varepsilon = 8$) | 67.90 | 70.56 | **0.466** | 8.669 | 2.370 |

**Attribution vs. DP-AdamW.** FIBER and DiSK both use two-point gradients, while DP-AdamW uses one-point. Thus, the FIBER vs. DP-AdamW gap reflects both the two-point method and FIBER's unique features. To isolate FIBER's contribution, we focus on comparisons with DiSK.

### 5.3. Ablation Studies

We ablate each FIBER component to quantify its contribution under matched privacy budgets and a consistent experimental protocol. We further analyze the compute–accuracy trade-off and empirically validate the variance attenuation factor $A(\omega)$ using a paired-run differencing procedure.

*Table 4.* Residual filtering vs. gradient-state exponential smoothing under the same training protocol and privacy budgets.

| $\varepsilon$ | Filter type | Acc. (%) | $\kappa$ | $\gamma$ |
|---|---|---|---|---|
| 0.5 | DiSK | 59.70 | 0.7 | 0.5 |
| 0.5 | FIBER | **65.32** | 0.6 | 0.7 |
| 2.0 | DiSK | 68.80 | 0.7 | 0.5 |
| 2.0 | FIBER | **71.39** | 0.6 | 0.7 |
| 8.0 | DiSK | **74.90** | 0.7 | 0.5 |
| 8.0 | FIBER | 73.19 | 0.6 | 0.7 |

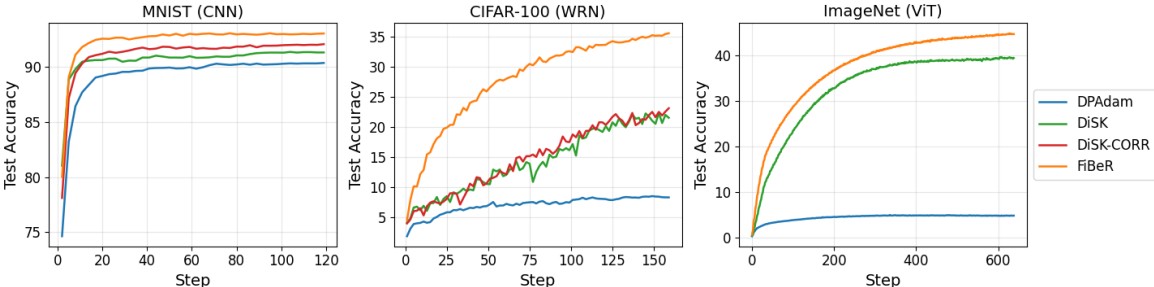

*Figure 2.* **Training dynamics at fixed privacy budgets.** Test accuracy curves on MNIST ($\varepsilon$=0.5), CIFAR-100 ($\varepsilon$=1), and ImageNet-1k ($\varepsilon$=8). MNIST and CIFAR-100 include DPAdam, DiSK, DiSK-CORR, and FIBER; ImageNet-1k includes DPAdam, DiSK, and FIBER.

**Innovation filtering vs. gradient-state smoothing.** We compare DiSK's gradient-state smoothing and FIBER's innovation filtering in a controlled setup. Both use the same DP mechanism and two-point construction; to match degrees of freedom, we tie FIBER's innovation gain to the two-point parameter ($\omega = \kappa$) and tune ($\kappa, \gamma$) pair for both. As shown in Table 4, innovation filtering benefits most in high-noise regimes: at $\varepsilon = 0.5$ and 2, FIBER outperforms DiSK by +5.20 and +2.59 points, respectively. When privacy is looser ($\varepsilon = 8$), the difference vanishes and DiSK is marginally better, suggesting the benefit of innovation filtering decreases as DP noise lessens. To isolate filtering from filter-aware correction, we disable the correction in FIBER by setting $A(\omega) = 0$.

**Effect of the two-point construction.** To isolate the contribution of the lookahead gradient evaluation, an additional evaluation is conducted a one-point variant of FIBER that uses a standard single-point DP gradient while retaining both the innovation filter and the filter-aware second-moment correction. Both components remain well-defined under single-point gradients. Therefore, the gap between this variant and the full FIBER method reflects the benefit of the two-point observation. As shown in Table 5, the one-point variant improves over DP-AdamW by only +2.64 points on average, whereas the full FIBER method achieves an improvement of +18.38 points. Removing the lookahead evaluation therefore reduces the average gain by 15.74 points. This result indicates that the two-point construction is a major contributor to the performance of FIBER, rather than merely serving as an implementation detail.

**Decoupling** ($\kappa, \gamma$) **from innovation gain** $\omega$. We test whether the temporal denoising gain $\omega$ provides an independent knob beyond the two-point geometry. Fixing ($\kappa, \gamma$) = $(0.6, 0.7)$, we sweep $\omega$ (innovation denoising of $\nu_t = g_t - \tilde{g}_{t-1}$) and observe a clear optimum near $\omega = 0.9$ in Table 6 (CNN5/CIFAR-10, $\varepsilon = 1$): small $\omega$ under-denoises, while $\omega \to 1$ weakens effective averaging and passes more noise. To further separate geometry from denoising, we fix $\gamma = 0.7$ and sweep ($\kappa, \omega$). Figure 3 (CNN5,

*Table 5.* Effect of removing the two-point construction on CNN5/CIFAR-10 under DP. The one-point FIBER variant keeps the innovation filter and filter-aware second-moment correction but replaces the two-point observation with a standard single-point DP gradient. $\Delta$ denotes the average improvement over DP-AdamW.

| Method | $\epsilon = 0.5$ | $\epsilon = 1$ | $\epsilon = 2$ | $\epsilon = 4$ | $\epsilon = 8$ | Avg. $\Delta$ |
|---|---|---|---|---|---|---|
| DP-AdamW | 44.77 | 49.77 | 54.61 | 60.52 | 63.59 | – |
| 1-pt FIBER | 46.67 | 53.39 | 56.61 | 63.32 | 66.49 | +2.64 |
| DiSK | 60.29 | 65.22 | 69.28 | 72.75 | 75.07 | +13.87 |
| Full FIBER | **66.92** | **70.91** | **74.40** | **75.96** | **76.99** | +18.38 |

$\varepsilon = 4$, 80 epochs) shows a smooth landscape where increasing $\omega$ from 0.5 to 0.9 improves accuracy for most $\kappa$, with the best setting again at ($\kappa, \omega$) = $(0.6, 0.9)$ (75.44%). In contrast, larger $\kappa$ degrades accuracy even at high $\omega$, indicating that ($\kappa, \gamma$) set the stable region while $\omega$ can be tuned within it. See Appendix E.5 for additional results.

*Table 6.* Decoupling the innovation gain $\omega$ from the two-point parameters ($\kappa, \gamma$).

| $\omega$ | 0.60 | 0.70 | 0.80 | 0.90 | 0.99 |
|---|---|---|---|---|---|
| Test Acc. (%) | 68.86 | 69.66 | 70.74 | **70.76** | 70.62 |

**Filter-aware second-moment correction.** To isolate the effect of filter-aware correction on AdamW's second moment, we compare: (i) full FIBER, which subtracts attenuated variance $A(\omega)\sigma_w^2$ from the bias-corrected second moment; (ii) a variant without subtraction (FIBER_NO_CORR); and (iii) a baseline (FIBER_BC_CORR) using the bias-correction strategy from Tang et al. (2024), which ignores filter attenuation. For fairness, we re-tune DiSK-CORR rather than reusing the optimal $\omega$ found for DiSK: we sweep $\omega$ (which jointly controls smoothing, the two-point geometry, and the attenuation factor $A(\omega)$) under the same tuning budget, and report DiSK-CORR using its own best-performing $\omega$. Figure 4: FIBER achieves the highest test accuracy across privacy budgets, outperforming both variants. This shows that adjusting for filter-altered DP noise in AdamW's second moment avoids preconditioner inflation and improves DP stability.

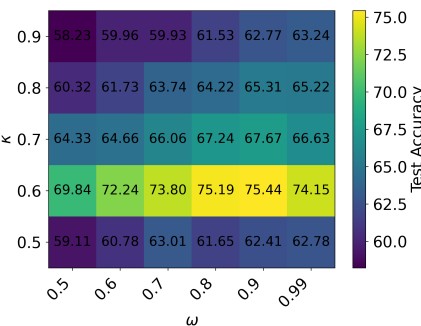

*Figure 3.* Sweep over $(\kappa, \omega)$ with fixed $\gamma = 0.7$.

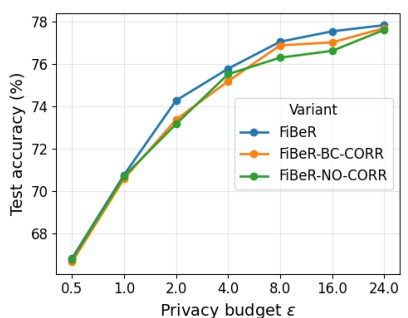

*Figure 4.* Test accuracy versus privacy budget $\varepsilon$ (CIFAR-10).

**Compute-Accuracy Tradeoff.** FIBER and DiSK use two-point gradient observations, requiring two backward passes per update; DP-AdamW uses a single-point update. Thus, per-update compute for FIBER and DiSK is at most twice that of DP-AdamW, though actual slowdowns are often smaller. Figure 5 shows FIBER achieves the best accuracy while remaining time-competitive and compares test accuracy at matched compute.

**Empirical check of variance attenuation.** To validate the theoretical attenuation factor $A(\omega)$, we run two replicas with identical initialization and minibatch order but independent DP noise. Define $\Delta g_t$ and $\Delta \tilde{g}_t$ as the differences between privatized and filtered gradients, respectively. We track $\rho_t = \mathrm{Var}(u^\top \Delta \tilde{g}_t)/\mathrm{Var}(u^\top \Delta g_t)$ using fixed random projections $u$. To ensure $\Delta g_t$ is noise-dominated, we report $r_t = \mathrm{Var}(u^\top \Delta g_t)/(2\sigma_w^2)$, which remains near 1. Figure 6 shows $\rho_t$ matches $A(\omega)$, supporting the attenuation model.

### 5.4. Diagnostic Studies

We validate FIBER through controlled diagnostics isolating (i) drift-tracking behavior and (ii) assumption compliance. Full methodology and results in Appendices E.3 and E.4.

**Conditions where innovation filtering helps.** We isolate drift-tracking by testing filters on synthetic gradients: constant-velocity (CV) for drift and random-walk (RW) for

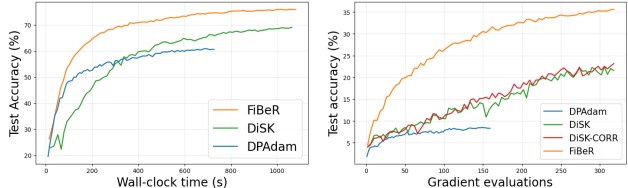

*Figure 5.* **Compute fairness diagnostics.** (a) Eval accuracy vs. wall-clock time on CIFAR-10 (CNN5, $\varepsilon = 4$). (b) Test accuracy vs. number of gradient evaluations on CIFAR-100 (WRN, $\varepsilon = 1$).

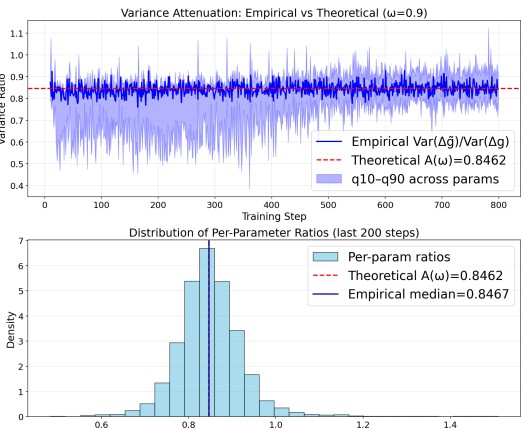

*Figure 6.* **Empirical validation of variance attenuation.** Top: $\rho_t = \mathrm{Var}(\Delta \tilde{g}_t)/\mathrm{Var}(\Delta g_t)$ during training for $\omega = 0.9$. Bottom: distribution of per-projection ratios over the last 200 steps.

stationary signals, both with DP noise. We report win rate, i.e., the number of trials (out of 7 random seeds) where innovation filtering achieves higher final utility than EMA state smoothing under the same $(\varepsilon, \delta)$ and training protocol. Table 7 shows innovation filtering wins consistently on CV (7/7 at $\varepsilon = 8$), while providing little benefit on RW.

*Table 7.* Synthetic drift: innovation filter win rates.

| Model | $\varepsilon = 2$ | $\varepsilon = 4$ | $\varepsilon = 8$ |
|-------|------|------|------|
| CV | 5/7 | 6/7 | 7/7 |
| RW | 0/7 | 0/7 | 1/7 |

**Correction validation.** We decompose the privatized gradient as $g_t = s_t + n_t$, where $s_t$ is the underlying clipped gradient signal and $n_t$ is the injected DP Gaussian noise (after filtering). Our variance subtraction is exact when $\mathbb{E}[s_t n_t] = 0$, in which case $\mathbb{E}[g_t^2] = \mathbb{E}[s_t^2] + \mathbb{E}[n_t^2]$ elementwise. Paired-run diagnostics on CIFAR-10 show projected correlation $\hat{\rho}(s, n) = 0.14$ and cross-term ratio $|\hat{\mathbb{E}}[sn]|/\hat{\mathbb{E}}[n^2] = 0.15$ – modest violations consistent with effective correction (Table 8). To prevent over-correction, we monitor clamp_mass$_t$ (fraction of $|\hat{m}_t|$ on floor-clamped coordinates). Figure 7 confirms small $\epsilon_v \leq 10^{-7}$ preserves adaptivity (clamp_mass < 0.01), while large $\epsilon_v \geq 10^{-6}$

*Table 8.* Proxy diagnostics for the filter-aware correction assumptions from a representative run (single random projection, index 0), reported over the steady-state phase.

| Metric | Value |
|---|---|
| $\omega$ | 0.9 |
| Total steps $T$ | 800 |
| Warmup / steady-state steps | 100 / 700 |
| $\widehat{\rho}(s, n)$ | 0.1404 |
| Cross-term ratio $\left\|\widehat{\mathbb{E}}[sn]\right\| / \widehat{\mathbb{E}}[n^2]$ | 0.1505 |
| Coeff. of variation (CV) | 0.0313 |

causes uniform preconditioning (clamp_mass $> 0.8$) ( See Appendix E.6 for more details). We set $\epsilon_v = 10^{-8}$ by default, which preserves adaptivity while preventing numerical instability from the variance subtraction.

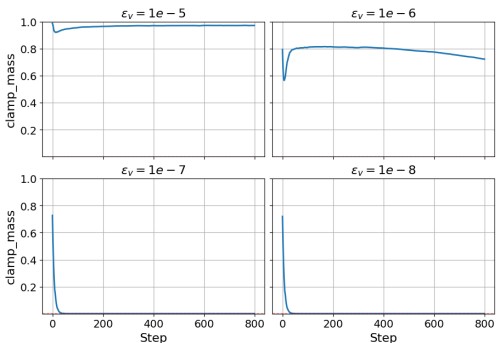

*Figure 7.* **Variance floor sensitivity.** Clamp diagnostics for $\epsilon_v \in \{10^{-8}, 10^{-7}, 10^{-6}, 10^{-5}\}$. Small floors ($10^{-8}, 10^{-7}$) yield negligible clamp activity after a short transient, while larger floors ($10^{-6}, 10^{-5}$) induce a floor-dominated regime.

## 6. Conclusion

Differential privacy introduces stochastic perturbations that can substantially degrade the behavior of adaptive optimizers. We presented FIBER, which (i) denoises privatized gradients in innovation space via a stable second-order recursion, (ii) decouples the two-point observation geometry from the temporal denoising gain for simpler tuning, and (iii) applies a filter-aware calibration to AdamW's second-moment estimator to match the post-filter noise statistics. Across vision and language benchmarks, FIBER consistently improves utility under fixed privacy budgets, with the largest gains in tighter-privacy and long-horizon regimes.

## Acknowledgements

This work was supported by the National Research Foundation of Korea (NRF) grant funded by the Korea government (MSIT) (RS-2025-00573160) and the "Advanced GPU Utilization Support Program" funded by the Government of the Republic of Korea (Ministry of Science and ICT). HLN was supported by the U.S. National Science Foundation grant CCF-2311649.

## Impact Statement

This paper presents work whose goal is to advance the field of Machine Learning. There are many potential societal consequences of our work, none which we feel must be specifically highlighted here.

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

## A. Differentially Private Optimization

### A.1. Differentially Private Optimization

We briefly recall differential privacy and the standard privatization mechanism used in DP optimization.

**Definition A.1** (($\varepsilon, \delta$)-Differential Privacy (Dwork & Roth, 2014)). A randomized mechanism $\mathcal{M}$ is ($\varepsilon, \delta$)-differentially private if for any two neighboring datasets $\mathcal{D}, \mathcal{D}'$ (differing in one example) and any measurable set $S$,

$$\Pr[\mathcal{M}(\mathcal{D}) \in S] \leq e^\varepsilon \Pr[\mathcal{M}(\mathcal{D}') \in S] + \delta. \tag{20}$$

**Definition A.2** (Gaussian Mechanism (Dwork & Roth, 2014; Wang et al., 2019)). Let $A : \mathcal{D} \rightarrow \mathbb{R}^d$ have $\ell_2$ sensitivity $\Delta_2(A) = \max_{\mathcal{D}, \mathcal{D}'} \|A(\mathcal{D}) - A(\mathcal{D}')\|_2$ over neighboring datasets. Then releasing $A(\mathcal{D}) + w$ with $w \sim \mathcal{N}(0, \sigma^2 \Delta_2(A)^2 I_d)$ is ($\varepsilon, \delta$)-DP for an appropriate choice of $\sigma$ as a function of ($\varepsilon, \delta$).

**DP-SGD.** DP-SGD applies the Gaussian mechanism to clipped per-example gradients. With $\mathrm{clip}(u, C) = u \cdot \min\{1, C/\|u\|_2\}$, the privatized minibatch gradient is

$$g_t = \frac{1}{B} \sum_{\xi \in B_t} \mathrm{clip}(\nabla f(\theta_t; \xi), C) + w_t, w_t \sim \mathcal{N}(0, \sigma_w^2 I_d),$$

where we parameterize noise by the noise multiplier $\sigma_{\mathrm{DP}}$ and set $\sigma_w = (\sigma_{\mathrm{DP}} C)/B$. Alternatively, noise $\mathcal{N}(0, (\sigma_{\mathrm{DP}} C)^2 I_d)$ can be added to the sum of clipped gradients, followed by division by $B$. The pseudocode for DP-SGD is presented in Appendix D.

**Privacy guarantee.** For sampling rate $q = B/|\mathcal{D}|$ and $T$ steps, DP-SGD is ($\varepsilon, \delta$)-DP when $\sigma_{\mathrm{DP}}$ is chosen using a standard privacy accountant.

## B. From Residual Gradient Dynamics to Innovation Filtering

This appendix completes the proof of Proposition 4.1 (Section 4.1). The derivation follows the pipeline

**(i) residual-gradient dynamics** $\Rightarrow$ **(ii) Kalman filter /** $\alpha$–$\beta$ **form** $\Rightarrow$ **(iii) simplifications** $\Rightarrow$ **FIBER**

We also note that if the drift state is removed (a random-walk model), the resulting filter reduces to a first-order exponential smoother (EMA), whereas the constant-velocity model below yields a second-order recursion.

**Conventions.** We present the derivation for a single coordinate (scalar) and suppress coordinate indices. Under a diagonal/coordinate-wise approximation, the same derivation applies element-wise to vectors. Throughout, $g_t$ denotes the observed (privatized) gradient at step $t$.

### B.1. Residual-Gradient Dynamics: Constant-Velocity Model

We model the latent (noise-free) gradient signal $s_t$ as evolving with a persistent drift $r_t$:

$$s_t = s_{t-1} + r_{t-1} + \eta_t, \tag{21}$$
$$r_t = r_{t-1} + \zeta_t, \tag{22}$$
$$g_t = s_t + w_t, \tag{23}$$

where $w_t$ is observation noise (including DP noise and minibatch noise), and $\eta_t, \zeta_t$ are process noises capturing model mismatch and changes in the latent gradient dynamics. Equations (21)-(23) correspond to the classical constant-velocity tracking model.

### B.2. Kalman Filter and the $\alpha$–$\beta$ Form

Define the 2D state $x_t \triangleq [\, s_t, \; r_t \,]^\top$. Then (21)-(23) becomes

$$x_t = A x_{t-1} + \varepsilon_t, \qquad g_t = H x_t + w_t, \tag{24}$$

where $\varepsilon_t = [\eta_t, \ \zeta_t]^\top$, and

$$A = \begin{bmatrix} 1 & 1 \\ 0 & 1 \end{bmatrix}, \qquad H = \begin{bmatrix} 1 & 0 \end{bmatrix}.$$

Let $\hat{x}_{t|t-1}$ and $\hat{x}_t$ denote the predicted and corrected estimates with covariances $P_{t|t-1}$ and $P_t$. The Kalman recursion is

$$\hat{x}_{t|t-1} = A\hat{x}_{t-1}, \tag{25}$$

$$P_{t|t-1} = AP_{t-1}A^\top + Q, \tag{26}$$

$$K_t = P_{t|t-1}H^\top \left( HP_{t|t-1}H^\top + R \right)^{-1}, \tag{27}$$

$$e_t = g_t - H\hat{x}_{t|t-1}, \tag{28}$$

$$\hat{x}_t = \hat{x}_{t|t-1} + K_t e_t, \tag{29}$$

$$P_t = (I - K_t H)P_{t|t-1}. \tag{30}$$

Since $HP_{t|t-1}H^\top$ is scalar, (27) only requires scalar division.

**$\alpha$–$\beta$ form.** Write $K_t = [\alpha_t, \ \beta_t]^\top$ and expand (25)-(29):

$$\hat{s}_{t|t-1} = \hat{s}_{t-1} + \hat{r}_{t-1}, \qquad\qquad\qquad \hat{r}_{t|t-1} = \hat{r}_{t-1}, \tag{31}$$

$$e_t = g_t - \hat{s}_{t|t-1} = g_t - (\hat{s}_{t-1} + \hat{r}_{t-1}), \tag{32}$$

$$\hat{s}_t = \hat{s}_{t|t-1} + \alpha_t e_t, \qquad\qquad\qquad \hat{r}_t = \hat{r}_{t|t-1} + \beta_t e_t. \tag{33}$$

This is the classical $\alpha$–$\beta$ filter for constant-velocity tracking.

### B.3. From Kalman Recursion to the FIBER Innovation Filter

We now apply standard simplifications to obtain the lightweight recursion used by FIBER.

**Step 1: Coordinate-wise (diagonal) approximation.** We treat coordinates independently (equivalently assume diagonal covariances), avoiding dense-matrix storage or inversion.

**Step 2: Steady-state constant gains.** When noise statistics are approximately stationary, Kalman gains converge quickly. We therefore replace time-varying gains by constants:

$$\alpha_t \approx \alpha, \qquad \beta_t \approx \beta. \tag{34}$$

**Step 3: Tied-gain reduction and identification with optimizer states.** To minimize tuning and match the optimizer implementation, we tie the gains:

$$\alpha = \beta = \omega \in (0, 1]. \tag{35}$$

(We emphasize that in the exact steady-state Kalman filter one generally has $\alpha_\infty \neq \beta_\infty$; the tied-gain choice (35) is a practical simplification.) We identify the Kalman estimates with the optimizer variables:

$$\tilde{g}_t \equiv \hat{s}_t, \qquad r_t \equiv \hat{r}_t. \tag{36}$$

Substituting (35)-(36) into (32)-(33) gives

$$r_t = r_{t-1} + \omega(g_t - (\tilde{g}_{t-1} + r_{t-1})) = (1 - \omega)r_{t-1} + \omega(g_t - \tilde{g}_{t-1}), \tag{37}$$

$$\tilde{g}_t = \tilde{g}_{t-1} + r_{t-1} + \omega(g_t - (\tilde{g}_{t-1} + r_{t-1})) = \tilde{g}_{t-1} + r_t. \tag{38}$$

**Residual-filter form.** Define the residual signal

$$\nu_t \triangleq g_t - \tilde{g}_{t-1}. \tag{39}$$

Then (37)-(38) becomes

$$r_t = (1 - \omega)r_{t-1} + \omega\nu_t, \tag{40}$$

$$\tilde{g}_t = \tilde{g}_{t-1} + r_t, \tag{41}$$

which matches (5)-(7) after identifying $\nu_t$ with the main-text notation.

**Second-order form.** Eliminating $r_t$ via $r_t = \tilde{g}_t - \tilde{g}_{t-1}$ yields

$$\tilde{g}_t = (2 - 2\omega)\tilde{g}_{t-1} - (1 - \omega)\tilde{g}_{t-2} + \omega g_t, \tag{42}$$

showing innovation filtering is second-order (two poles), unlike EMA.

## B.4. Convergence of Time-Varying Kalman Gains

The Kalman gain $K_t = [\alpha_t, \beta_t]^\top$ depends on the covariance $P_{t|t-1}$, which evolves according to a Riccati recursion. Under time-invariant $(A, H, Q, R)$ with $R > 0$, standard Kalman filtering theory implies that the covariance converges to a unique stabilizing fixed point (under detectability/stabilizability conditions), and therefore the gains converge to constants. For completeness, we now write the scalar recursion for the covariance entries.

**Noise model.** We take $Q = \text{diag}(\sigma_s^2, \sigma_r^2)$ and $R = \sigma_w^2$ in (26) and (27).

**Covariance parameterization.** Write the posterior covariance as

$$P_t = \begin{bmatrix} p_t & c_t \\ c_t & q_t^P \end{bmatrix}, \tag{43}$$

where $p_t = \text{Var}(s_t - \hat{s}_t)$, $q_t^P = \text{Var}(r_t - \hat{r}_t)$, and $c_t = \text{Cov}(s_t - \hat{s}_t, \ r_t - \hat{r}_t)$. Let $P_t^-$ denote the predicted covariance $P_{t|t-1}$ with the same parameterization $P_t^- = \begin{bmatrix} p_t^- & c_t^- \\ c_t^- & q_t^{P-} \end{bmatrix}$.

**Prediction.** From $P_t^- = AP_{t-1}A^\top + Q$, we obtain

$$p_t^- = p_{t-1} + 2c_{t-1} + q_{t-1}^P + \sigma_s^2, \tag{44}$$
$$c_t^- = c_{t-1} + q_{t-1}^P, \tag{45}$$
$$q_t^{P-} = q_{t-1}^P + \sigma_r^2. \tag{46}$$

**Gains.** The innovation variance is $S_t = HP_t^- H^\top + R = p_t^- + \sigma_w^2$, and the Kalman gain is

$$\alpha_t = \frac{p_t^-}{p_t^- + \sigma_w^2}, \qquad \beta_t = \frac{c_t^-}{p_t^- + \sigma_w^2}. \tag{47}$$

**Correction.** Using $P_t = (I - K_t H)P_t^-$ yields

$$p_t = (1 - \alpha_t)\, p_t^- = \frac{\sigma_w^2}{p_t^- + \sigma_w^2}\, p_t^-, \tag{48}$$

$$c_t = (1 - \alpha_t)\, c_t^- = \frac{\sigma_w^2}{p_t^- + \sigma_w^2}\, c_t^-, \tag{49}$$

$$q_t^P = q_t^{P-} - \beta_t c_t^- = q_t^{P-} - \frac{(c_t^-)^2}{p_t^- + \sigma_w^2}. \tag{50}$$

Equations (44)-(50) define a discrete-time Riccati recursion.

**Steady state and constant-gain approximation.** Under standard conditions with $\sigma_w^2 > 0$, the recursion converges to a stabilizing fixed point $(p_\infty, c_\infty, q_\infty^P)$, implying

$$(\alpha_t, \beta_t) \to (\alpha_\infty, \beta_\infty). \tag{51}$$

Empirically, this convergence is fast (a few tens of steps), which motivates replacing time-varying gains by constants and using the single tunable innovation gain $\omega$ in FIBER. Figure 8 visualizes this convergence on a representative setting.

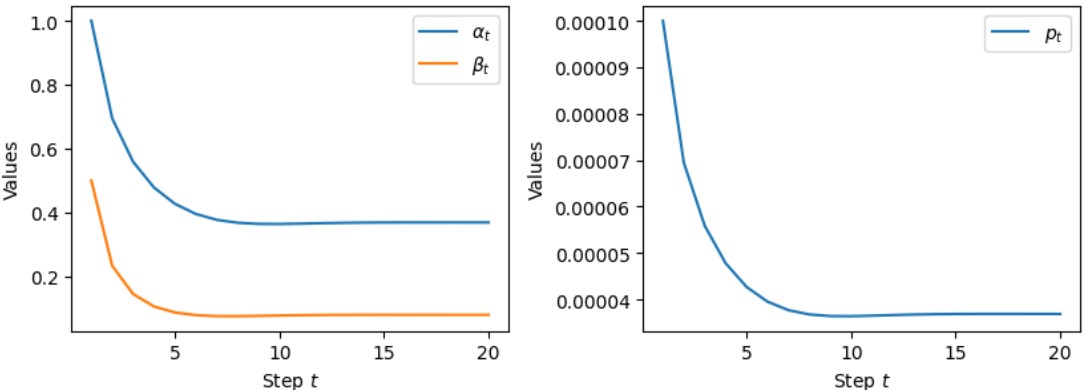

*Figure 8.* Convergence of $(\alpha_t, \beta_t)$ and $p_t^-$ under the constant-velocity Kalman recursion (Eqs. (44)-(50)), illustrating the rapid approach to steady-state gains.

## B.5. Comparison: DiSK Random-Walk Dynamics Gives EMA

For reference, DiSK-style gradient-state filtering corresponds to a random-walk latent gradient model

$$s_t = s_{t-1} + q_t, \qquad g_t = s_t + w_t \tag{52}$$

whose steady-state scalar Kalman filter reduces to the first-order EMA

$$\tilde{g}_t = (1 - \kappa)\tilde{g}_{t-1} + \kappa g_t. \tag{53}$$

Thus, FIBER differs at the modeling level (constant-velocity vs. random-walk), which leads to innovation filtering rather than gradient-state EMA smoothing.

## C. Filter-Aware Bias Correction

This appendix derives the DP noise attenuation factor $A(\omega)$ used in our filter-aware second-moment correction, and then connects it to AdamW's second-moment estimator.

### C.1. Noise Propagation Through the Innovation Filter

We work coordinate-wise (the filter is linear and DP noise is isotropic), and drop the coordinate index. Assume the innovation filter input is pure DP noise:

$$g_t = w_t, \qquad \mathbb{E}[w_t] = 0, \qquad \mathrm{Var}(w_t) = \sigma_w^2, \qquad w_t \perp w_{t'} \ (t \neq t'). \tag{54}$$

Recall the innovation recursion (main text (5)-(7)):

$$\nu_t = g_t - \tilde{g}_{t-1}, \qquad r_t = (1 - \omega)r_{t-1} + \omega\nu_t, \qquad \tilde{g}_t = \tilde{g}_{t-1} + r_t. \tag{55}$$

Substituting $g_t = w_t$ into (55) gives the linear system

$$\tilde{g}_t = (1 - \omega)\tilde{g}_{t-1} + (1 - \omega)r_{t-1} + \omega w_t, \tag{56}$$
$$r_t = -\omega\tilde{g}_{t-1} + (1 - \omega)r_{t-1} + \omega w_t. \tag{57}$$

Define the 2D filter state $z_t \triangleq [\tilde{g}_t, \ r_t]^\top$. Then

$$z_t = M z_{t-1} + \omega b\, w_t, \qquad M \triangleq \begin{bmatrix} 1 - \omega & 1 - \omega \\ -\omega & 1 - \omega \end{bmatrix}, \qquad b \triangleq \begin{bmatrix} 1 \\ 1 \end{bmatrix}. \tag{58}$$

**Stability.** The eigenvalues of $M$ have magnitude $\sqrt{1 - \omega}$, hence $M$ is Schur-stable for $\omega \in (0, 1]$. Therefore, a unique stationary covariance exists.

## C.2. Deriving $A(\omega)$ via a Lyapunov Equation

Let $\Sigma \triangleq \lim_{t \to \infty} \mathbb{E}[z_t z_t^\top]$ denote the stationary covariance. Since $w_t$ is independent of $z_{t-1}$, $\Sigma$ satisfies the discrete Lyapunov equation

$$\Sigma = M\Sigma M^\top + Q, \qquad Q \triangleq \omega^2 \sigma_w^2 \, bb^\top = \omega^2 \sigma_w^2 \begin{bmatrix} 1 & 1 \\ 1 & 1 \end{bmatrix}. \tag{59}$$

Write

$$\Sigma = \begin{bmatrix} x & y \\ y & z \end{bmatrix}, \qquad \text{so that } x = \operatorname{Var}(\tilde{g}_t), \ z = \operatorname{Var}(r_t). \tag{60}$$

Let $\rho \triangleq 1 - \omega$, so $M = \begin{bmatrix} \rho & \rho \\ -\omega & \rho \end{bmatrix}$. Expanding $M\Sigma M^\top$ and equating entries in (59) yields the linear system

$$x = \rho^2(x + 2y + z) + \omega^2 \sigma_w^2, \tag{61}$$

$$y = -\rho\omega(x + y) + \rho^2(y + z) + \omega^2 \sigma_w^2, \tag{62}$$

$$z = \omega^2 x - 2\rho\omega y + \rho^2 z + \omega^2 \sigma_w^2. \tag{63}$$

Solving (61)-(63) gives

$$x = \sigma_w^2 \frac{2 - \omega}{4 - 3\omega}, \tag{64}$$

$$y = \sigma_w^2 \frac{\omega}{4 - 3\omega}, \qquad z = \sigma_w^2 \frac{2\omega}{4 - 3\omega}. \tag{65}$$

Therefore,

$$\operatorname{Var}(\tilde{g}_t) = x = A(\omega)\sigma_w^2, \qquad A(\omega) \triangleq \frac{2 - \omega}{4 - 3\omega}. \tag{66}$$

For $\omega \in (0, 1]$, $A(\omega) \in [\frac{1}{2}, 1]$ with $A(1) = 1$ and $\lim_{\omega \to 0^+} A(\omega) = \frac{1}{2}$, proving Proposition 4.6.

## C.3. Implication for AdamW's Second Moment

AdamW forms the coordinate-wise second moment

$$v_t = \beta_2 v_{t-1} + (1 - \beta_2)\tilde{g}_t^2, \qquad \hat{v}_t = \frac{v_t}{1 - \beta_2^{t+1}}. \tag{67}$$

**Lemma C.1** (Expectation of Adam's bias-corrected second moment). *Let $v_t = \beta_2 v_{t-1} + (1 - \beta_2)\tilde{g}_t^2$ with $v_{-1} = 0$ and $\hat{v}_t = v_t/(1 - \beta_2^{t+1})$. Then for any process $\{\tilde{g}_t\}$,*

$$\mathbb{E}[\hat{v}_t] = \frac{1 - \beta_2}{1 - \beta_2^{t+1}} \sum_{k=0}^{t} \beta_2^{t-k} \, \mathbb{E}[\tilde{g}_k^2].$$

*In particular, if $\mathbb{E}[\tilde{g}_k^2] = m_2$ for all $k \leq t$, then $\mathbb{E}[\hat{v}_t] = m_2$ exactly.*

*Proof.* Unrolling the recursion yields $v_t = (1 - \beta_2) \sum_{k=0}^{t} \beta_2^{t-k} \tilde{g}_k^2$. Taking expectation and dividing by $(1 - \beta_2^{t+1})$ gives the result. $\qquad\square$

Under a standard local-stationarity approximation in which $\mathbb{E}[\tilde{g}_t^2]$ is (approximately) constant over the window of interest, unrolling (67) yields $\mathbb{E}[\hat{v}_t] \approx \mathbb{E}[\tilde{g}_t^2]$.

Now decompose $\tilde{g}_t = s_t + n_t$, where $n_t$ is the contribution of the filtered DP noise. By linearity of the filter and (66), $\mathbb{E}[n_t] = 0$ and $\mathbb{E}[n_t^2] = A(\omega)\sigma_w^2$ in steady state. Assuming $\mathbb{E}[s_t n_t] = 0$, we obtain

$$\mathbb{E}[\tilde{g}_t^2] = \mathbb{E}[s_t^2] + \mathbb{E}[n_t^2] = \mathbb{E}[s_t^2] + A(\omega)\sigma_w^2, \tag{68}$$

which is Proposition C.3. This motivates the filter-aware corrected preconditioner

$$\bar{v}_t = \max\big(\hat{v}_t - A(\omega)\sigma_w^2, \ \epsilon_v\big), \tag{69}$$

used by FIBER.

## C.4. Bias Correction for Gradient State Filter

We derive the DP noise attenuation factor for the gradient-state EMA used by DiSK. Consider the scalar recursion

$$\tilde{g}_t = (1 - \kappa)\tilde{g}_{t-1} + \kappa g_t, \tag{70}$$

and assume the input is pure i.i.d. DP noise $g_t = w_t$ with $\mathbb{E}[w_t] = 0$ and $\mathrm{Var}(w_t) = \sigma_w^2$. Unrolling (70) yields the stationary representation

$$\tilde{g}_t = \kappa \sum_{j=0}^{\infty} (1 - \kappa)^j w_{t-j}, \tag{71}$$

which is well-defined for $\kappa \in (0, 1]$. Using independence of $\{w_t\}$,

$$\mathrm{Var}(\tilde{g}_t) = \kappa^2 \sum_{j=0}^{\infty} (1 - \kappa)^{2j} \mathrm{Var}(w_{t-j}) = \kappa^2 \sigma_w^2 \sum_{j=0}^{\infty} (1 - \kappa)^{2j}$$

$$= \kappa^2 \sigma_w^2 \cdot \frac{1}{1 - (1 - \kappa)^2} = \sigma_w^2 \cdot \frac{\kappa}{2 - \kappa}. \tag{72}$$

Therefore, the EMA state filter attenuates i.i.d. DP noise by the factor

$$A_{\text{state}}(\kappa) \triangleq \frac{\mathrm{Var}(\tilde{g}_t)}{\sigma_w^2} = \frac{\kappa}{2 - \kappa} \in (0, 1], \tag{73}$$

with $A_{\text{state}}(\kappa) \to 0$ as $\kappa \to 0^+$ and $A_{\text{state}}(1) = 1$.

Analogous to Appendix C.3, this implies that when AdamW moments are computed from the EMA-filtered gradients, the expected DP noise contribution to the bias-corrected second moment is $A_{\text{state}}(\kappa)\sigma_w^2$. Thus, a filter-aware correction for DiSK is

$$\bar{v}_t^{(\text{state})} \triangleq \max\left(\hat{v}_t - A_{\text{state}}(\kappa)\sigma_w^2, \, \epsilon_v\right), \tag{74}$$

which we refer to as DiSK-CORR in the experiments.

## C.5. Finite-time DP noise bound for the innovation filter

**Lemma C.2** (Finite-time DP noise bound for the innovation filter). *Let $\tilde{w}_t$ denote the innovation-filter output driven by i.i.d. $w_t \sim \mathcal{N}(0, \sigma_w^2 I)$ and initialize $\tilde{g}_{-1} = r_{-1} = 0$. Then for all $t \geq 0$ and each coordinate $i$,*

$$\mathrm{Var}(\tilde{w}_{t,i}) \leq A(\omega)\sigma_w^2,$$

*and moreover $\mathrm{Var}(\tilde{w}_{t,i})$ converges to $A(\omega)\sigma_w^2$ at a geometric rate set by the spectral radius of the state matrix.*

*Proof.* Define the state $z_t \triangleq [\tilde{g}_t, \, r_t]^\top$. For noise-only input, the recursion can be written as the linear time-invariant system

$$z_t = M z_{t-1} + \omega b \, w_t, \qquad M \triangleq \begin{bmatrix} 1 - \omega & 1 - \omega \\ -\omega & 1 - \omega \end{bmatrix}, \quad b \triangleq \begin{bmatrix} 1 \\ 1 \end{bmatrix}.$$

Let $\Sigma_t \triangleq \mathbb{E}[z_t z_t^\top]$ denote the (uncentered) covariance since $\mathbb{E}[z_t] = 0$. Using independence of $w_t$ from $z_{t-1}$ and $\mathbb{E}[w_t^2] = \sigma_w^2$,

$$\Sigma_t = M \Sigma_{t-1} M^\top + Q, \qquad Q \triangleq \omega^2 \sigma_w^2 \, bb^\top,$$

with $\Sigma_{-1} = 0$.

By iterating the recursion, we obtain the closed form

$$\Sigma_t = \sum_{k=0}^{t} M^k Q (M^\top)^k,$$

which is positive semidefinite and nondecreasing in the Loewner order as $t$ increases. Since $\omega \in (0, 1]$, the matrix $M$ is Schur-stable (its eigenvalues have magnitude $\sqrt{1 - \omega} < 1$), so the infinite series converges to the unique stationary covariance

$$\Sigma_\infty = \sum_{k=0}^\infty M^k Q (M^\top)^k,$$

which is equivalently the unique solution to the discrete Lyapunov equation $\Sigma_\infty = M\Sigma_\infty M^\top + Q$.

Therefore $\Sigma_t \preceq \Sigma_\infty$ for all $t$, implying

$$\mathrm{Var}(\tilde{g}_t) = e_1^\top \Sigma_t e_1 \leq e_1^\top \Sigma_\infty e_1 = A(\omega)\sigma_w^2,$$

where $e_1 = [1, 0]^\top$. For the convergence rate, note that

$$\Sigma_\infty - \Sigma_t = \sum_{k=t+1}^\infty M^k Q (M^\top)^k = M^{t+1} \left( \sum_{j=0}^\infty M^j Q (M^\top)^j \right) (M^\top)^{t+1} = M^{t+1} \Sigma_\infty (M^\top)^{t+1}.$$

Hence, by submultiplicativity,

$$\|\Sigma_\infty - \Sigma_t\|_2 \leq \|M^{t+1}\|_2^2 \|\Sigma_\infty\|_2.$$

Since $M$ is Schur-stable ($\rho(M) = \sqrt{1 - \omega} < 1$), standard results on matrix powers imply that for any $\varepsilon > 0$ there exists a constant $C_\varepsilon$ such that $\|M^k\|_2 \leq C_\varepsilon (\rho(M) + \varepsilon)^k$ for all $k \geq 0$ (e.g., by Gelfand's formula). Therefore,

$$\|\Sigma_\infty - \Sigma_t\|_2 \leq C_\varepsilon^2 \|\Sigma_\infty\|_2 (\rho(M) + \varepsilon)^{2(t+1)},$$

which establishes geometric convergence. (For $\omega = 1$, $M$ is nilpotent and the convergence is finite-time.)

$\square$

### C.6. Second-moment decomposition and the cross term

**Proposition C.3** (Second-moment decomposition under filtered DP noise)**.** *Let $g_t = \bar{g}_t + w_t$, where $\{w_t\}$ are i.i.d. $\mathcal{N}(0, \sigma_w^2)$, and let $\tilde{g}_t$ be the output of a (fixed-initialization) stable linear filter $\mathcal{H}$ applied to $\{g_t\}$. Denote by $n_t \triangleq \mathcal{H}(w)_t$ the filter output when driven only by $\{w_t\}$, and define $s_t \triangleq \tilde{g}_t - n_t$ so that $\tilde{g}_t = s_t + n_t$. If the DP noise variance gain of $\mathcal{H}$ is $A$ in the sense that $\mathbb{E}[n_t^2] = A\sigma_w^2$ in steady state, then*

$$\mathbb{E}[\tilde{g}_t^2] = \mathbb{E}[s_t^2] + A\sigma_w^2 + 2\,\mathbb{E}[s_t n_t]. \tag{75}$$

*In particular, if $\mathbb{E}[s_t n_t] = 0$ (e.g., if $s_t$ is independent of $\{w_\tau\}_{\tau \leq t}$), then*

$$\mathbb{E}[\tilde{g}_t^2] = \mathbb{E}[s_t^2] + A\sigma_w^2. \tag{76}$$

*Proof.* By linearity of $\mathcal{H}$, $\tilde{g}_t = \mathcal{H}(g)_t = \mathcal{H}(\bar{g})_t + \mathcal{H}(w)_t$. With $n_t = \mathcal{H}(w)_t$ and $s_t = \tilde{g}_t - n_t$, we have $\tilde{g}_t = s_t + n_t$. Expanding the square and taking expectations gives $\mathbb{E}[\tilde{g}_t^2] = \mathbb{E}[s_t^2] + \mathbb{E}[n_t^2] + 2\mathbb{E}[s_t n_t]$. Substituting $\mathbb{E}[n_t^2] = A\sigma_w^2$ yields (75). If $\mathbb{E}[s_t n_t] = 0$, (76) follows. $\square$

## D. Algorithm Pseudocode

**DP-SGD**    Algorithm 2 shows the pseudocode for the DP-SGD optimizer.

**FIBER**    The pseudocode for the FIBER optimizer is presented in Algorithm 3.

## E. Additional Numerical Results

This appendix provides implementation details, hyperparameter choices, and additional experimental results.

---

**Algorithm 2** DP-SGD (per-example clipping + Gaussian noise) (Abadi et al., 2016)

---

1: **Input:** dataset $\mathcal{D}$, steps $T$, batch size $B$, clipping $C$, step sizes $\{\eta_t\}$, noise std $\sigma_w^2 \leftarrow (\sigma_{\text{DP}} C)^2 / B^2$ where $\sigma_{\text{DP}}$ is noise multiplier
2: Initialize $\theta_0$
3: **for** $t = 0$ **to** $T - 1$ **do**
4:     Sample minibatch $B_t \subset \mathcal{D}$ with $|B_t| = B$
5:     $\bar{g}_t \leftarrow \frac{1}{B} \sum_{\xi \in B_t} \text{clip}(\nabla f(\theta_t; \xi), C)$
6:     Sample $w_t \sim \mathcal{N}(0, \sigma_w^2 I_d)$;   $g_t \leftarrow \bar{g}_t + w_t$
7:     $\theta_{t+1} \leftarrow \theta_t - \eta_t g_t$
8: **end for**
9: **Output:** $\theta_T$

---

---

**Algorithm 3** FIBER: DP-AdamW with Innovation Filtering and Filter-Aware Bias Correction

---

1: **Input:** $\theta_0$, dataset $\mathcal{D}$, steps $T$, batch size $B$, lr $\eta$, clip $C$, DP noise $\sigma_{\text{DP}}$, Adam params $(\beta_1, \beta_2, \epsilon)$, weight decay $\lambda$, innovation gain $\omega$, floor $\epsilon_v$, (optional) two-point $(\kappa, \gamma)$.
2: **Init:** $\tilde{g}_{-1} = 0$, $r_{-1} = 0$, $m_{-1} = 0$, $v_{-1} = 0$, $d_{-1} = 0$, $\sigma_w^2 \leftarrow (\sigma_{\text{DP}} C)^2 / B^2$, $A(\omega) \leftarrow \frac{2-\omega}{4-3\omega}$.
3: **for** $t = 0, \ldots, T - 1$ **do**
4:     Sample minibatch $B_t \subset \mathcal{D}$, $|B_t| = B$.
5:     Privatized gradient observation:

$$g_t = \frac{1}{B} \sum_{\xi \in B_t} \text{clip}\Big(\frac{1-\kappa}{\kappa\gamma} \nabla f(\theta_t + \gamma d_{t-1}; \xi) + \Big(1 - \frac{1-\kappa}{\kappa\gamma}\Big) \nabla f(\theta_t; \xi), \, C\Big) + w_t, \quad w_t \sim \mathcal{N}(0, \sigma_w^2 I_d).$$

6:     Innovation filter: $\nu_t \leftarrow g_t - \tilde{g}_{t-1}$; $r_t \leftarrow (1 - \omega) r_{t-1} + \omega \nu_t$; $\tilde{g}_t \leftarrow \tilde{g}_{t-1} + r_t$.
7:     Adam moments: $m_t \leftarrow \beta_1 m_{t-1} + (1 - \beta_1) \tilde{g}_t$; $v_t \leftarrow \beta_2 v_{t-1} + (1 - \beta_2)(\tilde{g}_t \odot \tilde{g}_t)$.
8:     Bias + filter-aware correction: $\hat{m}_t \leftarrow m_t / (1 - \beta_1^{t+1})$, $\hat{v}_t \leftarrow v_t / (1 - \beta_2^{t+1})$, $\bar{v}_t \leftarrow \max(\hat{v}_t - A(\omega)\sigma_w^2, \epsilon_v)$.
9:     Update: $\theta_{t+1} \leftarrow (1 - \eta\lambda)\theta_t - \eta \hat{m}_t \oslash (\sqrt{\bar{v}_t} + \epsilon)$; $d_t \leftarrow \theta_{t+1} - \theta_t$.
10: **end for**
11: **Output:** $\theta_T$.

---

### E.1. Experiment Details

**Code and reproducibility.** All experiments are conducted in PyTorch (Paszke et al., 2019). FIBER is implemented as a drop-in optimizer that operates on privatized gradients from the same DP training pipeline as the baselines, which apply per-example gradient clipping and additive Gaussian noise. For a target $(\varepsilon, \delta)$ budget, we compute the required noise multiplier using a RDP accountant (Wang et al., 2019; Bu et al., 2023b), as implemented in standard DP libraries such as Opacus (Yousefpour et al., 2021) and FastDP (Bu et al., 2024b). Full code is available at `https://anonymous.4open.science/r/InnoAdamBC-4752`. Our implementation uses FastDP v2.1, Python 3.12, CUDA 12.6, and PyTorch 2.9.

**Hardware.** Unless otherwise specified, each trial is run on a single GPU. We use RTX 4090 (24GB) or RTX 5090 (32GB) for most benchmarks, and RTX Pro 6000 (96GB) for ImageNet-1k experiments. Training time varies with the dataset and model size; the most expensive setting is ViT-small training on ImageNet-1k, which completes in under 15 days.

**Tuning budget and fairness.** We allocate a fixed hyperparameter search budget of $N_{\text{total}} = 100$ trials per method for each dataset and privacy setting. Each method is tuned independently within this budget, and results are reported for the best configuration identified under these constraints. For FIBER, we employ a staged search-first selecting $(\kappa, \gamma)$, then tuning $\omega$-as a practical strategy for navigating the search space; importantly, this does not increase the total number of trials. The best configuration found within the fixed budget is reported for each method and setting.

**Training recipe.** Gradient accumulation is employed to support large effective batch sizes. For training from scratch, the learning rate is warmed up for a fixed fraction of total steps (e.g., $1/20$), followed by cosine decay. Model- and dataset-specific settings are kept consistent across methods, including data augmentation, normalization, and EMA when

applicable. To ensure reproducibility, the random seed is set to 42.

**Metrics.** For vision tasks, we report top-1 accuracy, using test accuracy for MNIST and CIFAR datasets and validation accuracy for ImageNet-1k, in accordance with established benchmark practices. For GLUE tasks, we report the official evaluation metric for each task: accuracy for MNLI, QNLI, and SST-2, and F1 score for QQP. For the E2E data, we report standard text-generation metrics including BLEU, ROUGE-L, METEOR, NIST, and CIDEr, as these metrics capture complementary aspects of output quality. These metrics assess various dimensions such as n-gram overlap, recall-oriented overlap, and consensus with reference texts. We use standard benchmark implementations and default settings for all metrics to ensure comparability with prior work.

### E.2. Hyperparameter Selection

**Main hyperparameters.** The primary hyperparameters include the number of epochs $E$, batch size $B$, learning rate $\eta$, and clipping threshold $C$, as well as the DP noise multiplier $\sigma_{\mathrm{DP}}$ computed by the accountant. FIBER introduces additional parameters, including innovation gain $\omega$, two-point parameters $(\kappa, \gamma)$, and a variance floor $\epsilon_v$ to ensure numerical stability in the filter-aware second-moment correction. Unless otherwise specified, we set $(\kappa, \gamma, \omega) = (0.6, 0.7, 0.9)$ as a default because our sensitivity analyses indicate this choice is near-optimal and robust across a broad range of settings (Appendix E.5). For CIFAR and MNIST, we follow prior work (Zhang et al., 2024a) and select hyperparameters on the test set due to the absence of an official validation split.

**Privacy accounting.** The noise multiplier is computed using an RDP accountant under fixed batch size sampling without replacement. We use RDP orders $\alpha \in \{1.1, 1.2, \ldots, 10.0, 12, 13, \ldots, 63\}$ and convert to $(\varepsilon, \delta)$ via the standard bound $\varepsilon = \min_\alpha \left( \epsilon_{\mathrm{RDP}}(\alpha) + \frac{\log(1/\delta)}{\alpha - 1} \right)$.

**Clipping and $\delta$.** Normalized $\ell_2$ clipping is applied as follows:

$$\mathrm{clip}(g, C) \triangleq g \cdot \min \left\{ 1, \frac{C}{\|g\|_2} \right\}. \tag{77}$$

Unless otherwise specified, $\delta$ is set to $1/N^{1.1}$. Table 9 provides the $\delta$ values for each dataset and task.

*Table 9.* Privacy parameter $\delta = 1/N^{1.1}$ used across datasets/tasks.

| Dataset/Task | $\delta$ |
|---|---|
| MNIST | $5.5 \times 10^{-6}$ |
| CIFAR-10 | $6.8 \times 10^{-6}$ |
| CIFAR-100 | $6.8 \times 10^{-6}$ |
| ImageNet-1k | $1.9 \times 10^{-7}$ |
| MNLI | $6.3 \times 10^{-7}$ |
| QNLI | $4.8 \times 10^{-7}$ |
| SST-2 | $4.9 \times 10^{-6}$ |
| QQP | $7.6 \times 10^{-7}$ |
| E2E | $1.2 \times 10^{-5}$ |
| DART | $6.1 \times 10^{-6}$ |

**Search grids.** Table 10 summarizes hyperparameter grids for vision training from scratch. AdamW defaults $(\beta_1, \beta_2) = (0.9, 0.999)$ and $\epsilon = 10^{-8}$ are used unless otherwise specified. $\epsilon_v = 10^{-8}$ is set by default. The detail reason for choosing this value of $\epsilon_v = 10^{-8}$ is discussed in Appendix E.6

### E.3. Synthetic Drift Diagnostics for Innovation Filtering

Table 4 indicates that innovation filtering can substantially help in some regimes (especially when gradients drift), yet it can be less favorable in other regimes where simpler EMA smoothing suffices (e.g., at larger $\varepsilon$ in Table 4). To provide direct evidence isolating drift-tracking effects from end-to-end network training, we run a controlled synthetic experiment where the latent (noise-free) gradient signal follows a known drift model, and the observed gradients are corrupted by additive

*Table 10.* Search grids for vision training-from-scratch (optimal values in bold). We tune $(E, B, \eta)$ first, then tune $(\omega, \kappa, \gamma)$ on top of the best base setup.

|  | MNIST | CIFAR | ImageNet-1k |
|---|---|---|---|
| Epochs $E$ | $\{1, \mathbf{2}, 3\} \times 20$ | $\{1, \mathbf{2}, 3, 4\} \times 40$ | $\{3, \mathbf{4}\} \times 40$ |
| Batch size $B$ | $\{2, \mathbf{5}\} \times 10^3$ | $\{0.5, 1, 2, \mathbf{5}\} \times 10^3$ | $\{\mathbf{5}, 10\} \times 10^3$ |
| LR $\eta$ | $\{0.5, \mathbf{0.25}, 0.05, 0.025\}$ | $\{\mathbf{0.005}, 0.001, 0.0005, 0.0001\}$ | $\{10, 3, 1, \mathbf{0.3}, 0.1\} \times 10^{-3}$ |
| $\omega$ | $\{\mathbf{0.9}\}$ | $\{0.6, 0.7, 0.8, \mathbf{0.9}, 0.99\}$ | $\{\mathbf{0.9}\}$ |
| $\kappa$ | $\{\mathbf{0.6}\}$ | $\{0.5, \mathbf{0.6}, 0.7, 0.8, 0.9\}$ | $\{\mathbf{0.6}\}$ |
| $\gamma$ | $\{\mathbf{0.7}\}$ | $\{0.6, \mathbf{0.7}, 0.8, 0.9\}$ | $\{\mathbf{0.7}\}$ |

Gaussian noise. We then compare (i) EMA smoothing (DiSK-style) and (ii) our innovation filter (Eqs. (5)–(7)) on their ability to track the latent signal.

**Generative models.** We generate a latent "true" gradient signal $\{s_t\}_{t=1}^T \subset \mathbb{R}^d$ and noisy observations $\{g_t\}$ via

$$g_t = s_t + w_t, \qquad w_t \sim \mathcal{N}(0, \sigma_w^2 I). \tag{78}$$

We consider two canonical drift models:

- **Constant-velocity (CV) model.** The latent signal exhibits trend-like dynamics (second-order drift), i.e., $s_t$ evolves with approximately constant velocity up to small perturbations.

- **Random-walk (RW) model.** The latent signal follows first-order diffusive dynamics, i.e., $s_t$ evolves as a random walk (near-stationary increments).

We summarize noise conditions using the signal-to-noise ratio (SNR) shown in Fig. 10:

$$\mathrm{SNR} \triangleq \sigma_s^2/\sigma_w^2, \tag{79}$$

where $\sigma_s^2$ denotes the latent signal scale (as instantiated in the synthetic generator) and $\sigma_w^2$ is the observation-noise variance.

**Filters compared.** Given observations $\{g_t\}$, we form filtered estimates $\{\tilde{g}_t\}$ using:

- **EMA (DiSK-style smoothing).** A first-order exponential moving average of the observed gradients.

- **Innovation filter (FIBER).** A constant-gain innovation recursion that filters the residual (innovation) stream and integrates it to form $\tilde{g}_t$ (Eqs. (5)–(7)).

**Evaluation metrics.** For each run, we evaluate tracking quality using the mean-squared error (MSE) against the latent signal:

$$\mathrm{MSE}(\tilde{g}, s) = \frac{1}{T} \sum_{t=1}^T \|\tilde{g}_t - s_t\|_2^2. \tag{80}$$

We report the relative improvement of innovation filtering over EMA as:

$$\mathrm{Improvement}(\%) \triangleq 100 \cdot \frac{\mathrm{MSE}_{\mathrm{EMA}} - \mathrm{MSE}_{\mathrm{Innov}}}{\mathrm{MSE}_{\mathrm{EMA}}}. \tag{81}$$

The percent improvement in Eq. (81) can be unbounded below when $\mathrm{MSE}_{\mathrm{EMA}}$ is small, meaning large negative values reflect rare but severe failures of innovation filtering in some regimes (particularly under RW dynamics). To provide a robust summary, we also report win rates and medians, and clip heatmap visualizations to $[-200\%, 100\%]$ for readability. Positive values indicate that innovation filtering improves tracking (i.e., lowers MSE relative to EMA). The **win rate** is defined as the fraction of evaluated configurations with positive improvement.

**Experiment sweep and privacy budgets.**    We run the synthetic diagnostic for privacy budgets $\varepsilon \in \{0.5, 1, 2, 4, 8\}$. For each $\varepsilon$, we evaluate a fixed set of 7 configurations (hence win rates are multiples of $1/7$), spanning a range of noise/drift conditions; the corresponding SNR coverage is visualized in Fig. 10. For readability in the heatmaps, we clip displayed improvements to $[-200\%, 100\%]$, while summary statistics (Table 11) are computed from the underlying (unclipped) values.

**Aggregate results across privacy budgets.**    Table 11 summarizes how the innovation filter behaves as the privacy budget varies. Under the CV model, the innovation filter's win rate increases monotonically with $\varepsilon$ (from $3/7$ at $\varepsilon = 0.5$ to $7/7$ at $\varepsilon = 8$), and the best-case improvement is consistently near $100\%$ across all privacy budgets. However, at very small $\varepsilon$ the average improvement can be dominated by rare catastrophic failures (large negative outliers), yielding a negative mean despite substantial best-case gains; as $\varepsilon$ increases, the average improvement becomes positive (e.g., $+59.9\%$ at $\varepsilon = 4$ and $+89.2\%$ at $\varepsilon = 8$). This pattern is consistent with innovation filtering being well-matched to trend-like (CV) dynamics, but requiring sufficient effective SNR and/or conservative gain settings to avoid instability when observation noise dominates.

In contrast, under the RW model, innovation filtering is rarely favorable: the win rate is $0/7$ for $\varepsilon \leq 4$, and only $1/7$ at $\varepsilon = 8$. The best-case improvement becomes less negative as $\varepsilon$ increases (from $-1225.6\%$ at $\varepsilon = 0.5$ to $-15.4\%$ at $\varepsilon = 4$), and becomes positive in a single configuration at $\varepsilon = 8$ ($+69.2\%$), but the mean remains strongly negative overall. These results align with the intuition that EMA smoothing is closer to the appropriate constant-gain estimator under RW-like dynamics, whereas innovation filtering is designed to track persistent trend (CV) behavior.

**Qualitative tracking behavior.**    Fig. 11 illustrates a representative CV run at $\mathrm{SNR} = 0.05$, plotting the $\ell_2$ tracking error over time. EMA error accumulates steadily, reflecting lag under drift, whereas innovation filtering maintains a low and approximately stable error over the entire horizon, directly visualizing the drift-tracking advantage in a regime where the model matches the innovation filter's inductive bias.

The synthetic diagnostic provides controlled support for the regime-dependent behavior seen in network training: innovation filtering is advantageous when gradients exhibit trend-like drift (CV) and the effective SNR is not extremely low, while EMA smoothing can be competitive or preferable under RW-like (diffusive) dynamics. We therefore interpret cases where DiSK outperforms at large $\varepsilon$ (e.g., $\varepsilon = 8$ in Table 4) as consistent with a regime where the gradient signal is closer to RW/stationary behavior and additional innovation dynamics are unnecessary.

*Table 11.* Synthetic drift summary across privacy budgets. "Win rate" is the fraction of configurations (out of 7) where innovation filtering improves over EMA (Eq. (81)). "Best" and "Avg" are the best-case and mean improvements (%).

| $\varepsilon$ | Model | Win rate (%) | Wins | Best (%) | Avg (%) |
|---|---|---|---|---|---|
| 0.5 | CV | 42.9 | 3/7 | 99.59 | -388.59 |
| 1.0 | CV | 57.1 | 4/7 | 99.90 | -189.32 |
| 2.0 | CV | 71.4 | 5/7 | 99.97 | -26.14 |
| 4.0 | CV | 85.7 | 6/7 | 99.99 | 59.91 |
| 8.0 | CV | 100.0 | 7/7 | 100.00 | 89.23 |
| 0.5 | RW | 0.0 | 0/7 | -1225.59 | -1466.996 |
| 1.0 | RW | 0.0 | 0/7 | -781.19 | -1396.070 |
| 2.0 | RW | 0.0 | 0/7 | -277.68 | -1297.283 |
| 4.0 | RW | 0.0 | 0/7 | -15.40 | -1180.544 |
| 8.0 | RW | 14.3 | 1/7 | 69.15 | -1014.144 |

### E.4. Empirical Assumption Audits

**Empirical audit of Proposition C.3 assumptions.**    The filter-aware correction in Proposition C.3 relies on the steady-state second-moment decomposition (Proposition C.3), which assumes the standard uncorrelatedness approximation $\mathbb{E}[s_{t,i} n_{t,i}] = 0$, and on a local-stationarity approximation in Appendix C.3 (treating $\mathbb{E}[\tilde{g}_{t,i}^2]$ as approximately constant over the averaging window). As also discussed in Section 6, these assumptions are approximate under closed-loop optimization, and may be affected by regimes where DP noise dominates or where clipping/noise is heterogeneous.

To quantify the approximation error, we extend the paired-run differencing protocol of Section 5.3 beyond variance attenuation. In addition to estimating the attenuation factor $A(\omega)$, we estimate a proxy for the cross term $\mathbb{E}[s_t n_t]$ via projected logging. Let $u \in \mathbb{R}^d$ be a fixed random unit vector shared across runs/projections. We log (i) the projected filtered gradient $x_t \triangleq u^\top \tilde{g}_t$ during training and (ii) the projected DP noise realization $y_t \triangleq u^\top w_t$ (available at generation time in

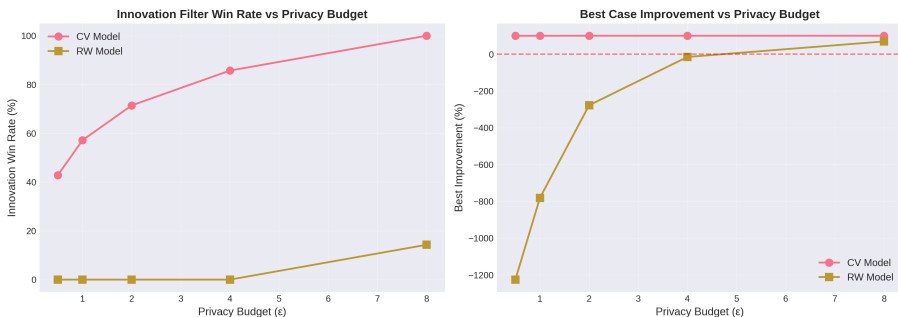

*Figure 9.* Synthetic drift diagnostic: (left) innovation filter win rate vs. privacy budget $\varepsilon$; (right) best-case improvement vs. $\varepsilon$, for both CV and RW latent dynamics.

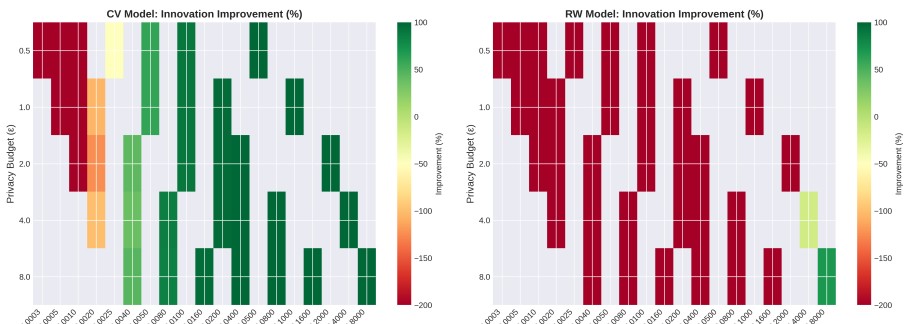

*Figure 10.* Improvement heatmaps (clipped to $[-200\%, 100\%]$ for visualization) across SNR values and privacy budgets $\varepsilon$, shown separately for CV (left) and RW (right) latent dynamics.

the code). We then apply the same innovation filter (Eqs. (5)–(7)) to the noise projection $\{y_t\}$ offline to obtain the projected filtered-noise component $n_t^{(\mathrm{proj})}$. Finally we define the projected signal proxy $s_t^{(\mathrm{proj})} \triangleq x_t - n_t^{(\mathrm{proj})}$ and compute summary statistics after a warmup period.

**Results.** Table 12 reports a representative diagnostic at $\omega = 0.9$ over $T = 800$ steps, using a warmup of 100 steps and statistics computed on the remaining 700 steps. We observe that the local-stationarity proxy holds well (coefficient of variation $\approx 0.031$ in a sliding-window variance estimate), supporting the use of a steady-state approximation over this window. Regarding uncorrelatedness, the projected correlation is modest ($\widehat{\rho}(s, n) \approx 0.140$), and the normalized cross term is small relative to the filtered-noise energy, $\left|\widehat{\mathbb{E}}[sn]\right|/\widehat{\mathbb{E}}[n^2] \approx 0.1505$. Overall, this diagnostic suggests that while closed-loop coupling is not strictly zero, its magnitude is limited in this setting and is consistent with the approximation used in Proposition C.3. This supports the limitations discussion in Section 6 and motivates reporting assumption-audit diagnostics alongside the correction.

**Practical implications and failure-mode guidance.** When the cross term is non-negligible, subtracting only $A(\omega)\sigma_w^2$ (Equation 12) may not fully remove optimizer-state inflation (if $\mathbb{E}[sn] > 0$) or may over-correct (if $\mathbb{E}[sn] < 0$). As a conservative safeguard, we recommend (i) reporting the assumption-audit metrics across multiple random projections and seeds, and (ii) monitoring over-correction diagnostics such as the "clamp mass" statistic in Appendix E.6. In regimes where the estimated cross-term ratio is consistently large, we suggest reducing the innovation gain $\omega$, increasing the variance floor $\epsilon_v$, or using layerwise clipping/noise as a robustness test.

### E.5. Hyperparameter Sensitivity

We investigate the sensitivity of FIBER to the two-point hyperparameters $(\kappa, \gamma)$ using the CNN5 model on the CIFAR-10 dataset. The experiments are conducted under a privacy budget of $\varepsilon = 4$ for 80 epochs. The innovation gain is fixed at $\omega = 0.9$, and a grid search is performed over $(\kappa, \gamma)$, while all other training and differential privacy settings remain constant.

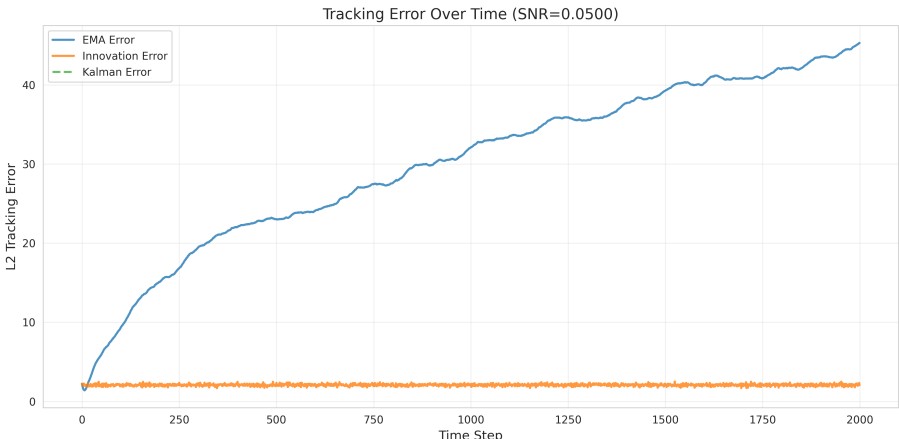

*Figure 11.* Representative CV run at $\mathrm{SNR} = 0.05$: $\ell_2$ tracking error over time for EMA vs. innovation filtering (and Kalman reference, if included in the script). Innovation filtering exhibits substantially lower drift-tracking error across the horizon.

*Table 12.* Proxy diagnostics for Proposition C.3 assumptions in one representative run (projection index 0). "Cross-term ratio" is $\left|\widehat{\mathbb{E}}[sn]\right|/\widehat{\mathbb{E}}[n^2]$.

| Metric | Value |
|---|---|
| $\omega$ | 0.9 |
| Total steps $T$ | 800 |
| Warmup / steady-state steps | 100 / 700 |
| $\widehat{\rho}(s, n)$ | 0.1404 |
| Cross-term ratio $\left|\widehat{\mathbb{E}}[sn]\right|/\widehat{\mathbb{E}}[n^2]$ | 0.1505 |
| Coeff. of variation (CV) | 0.0313 |

Figure 12(**a**) presents the resulting test accuracy as a function of $(\kappa, \gamma)$. Several notable trends are observed. First, performance varies smoothly across most of the grid, indicating that FIBER does not exhibit excessive sensitivity to moderate changes in $(\kappa, \gamma)$. Second, the optimal region is concentrated around $\kappa \approx 0.6$ and $\gamma \approx 0.7$, where the peak accuracy of 75.44% is observed. Deviation from this region generally results in reduced performance, particularly for larger $\kappa$ values (e.g., $\kappa \geq 0.8$), where accuracy plateaus in the mid-60% range. Third, very small $\kappa$ values can be unstable depending on $\gamma$. At $\kappa = 0.5$, the accuracy varies substantially (from 52.82% to 73.33%), indicating that overly aggressive two-point mixing may be sensitive when combined with particular extrapolation scales.

Recall that the two-point construction uses a mixing coefficient

$$a \triangleq \frac{1 - \kappa}{\kappa \gamma}, \tag{82}$$

so that (before clipping/noise) the two-point per-example vector is a weighted combination of a lookahead gradient and a current gradient. A natural stability requirement is that this combination remains convex, i.e., $0 \leq a \leq 1$, which avoids negative weights that can amplify the update norm and interact poorly with clipping. For $\kappa \in (0, 1)$ and $\gamma > 0$, we always have $a \geq 0$, and $a \leq 1$ is equivalent to the simple constraint

$$\gamma \geq \frac{1 - \kappa}{\kappa}. \tag{83}$$

This constraint is particularly meaningful under DP because clipping is nonlinear: if $a > 1$, the weight on the current gradient becomes negative, which can increase the norm of the combined vector and trigger additional clipping, thereby increasing clipping-induced distortion. Conversely, when $a \in [0, 1]$ (convex mixing), the combined vector cannot exceed the convex hull of the two endpoints, providing a basic guardrail against norm explosion prior to clipping.

The observed optimum $(\kappa, \gamma) \approx (0.6, 0.7)$ aligns closely with this convex-mixing boundary. Indeed, $\kappa = 0.6$ implies $(1 - \kappa)/\kappa \approx 0.667$, and $\gamma = 0.7$ lies just above this threshold, yielding $a \approx 0.4/(0.6 \cdot 0.7) \approx 0.952$. Therefore, the

optimal region corresponds to a near-lookahead update, in which most weight is assigned to the lookahead gradient while retaining a small stabilizing weight on the current gradient. This configuration is consistent with enhanced robustness under clipping and differential privacy noise. At the boundary $\gamma = (1 - \kappa)/\kappa$, we have $a = 1$, i.e., the construction becomes "pure lookahead"; slightly larger $\gamma$ keeps the mixture as convex ($a < 1$) while providing additional robustness.

Next, we study sensitivity with respect to the temporal denoising gain by fixing $\gamma = 0.7$ (near-optimal from the previous sweep) and tuning $(\kappa, \omega)$. Figure 12(**b**) demonstrates that accuracy generally improves as $\omega$ increases from 0.5 to 0.9 across most $\kappa$ values, which is consistent with the observation that stronger innovation smoothing is beneficial under stringent privacy constraints. The optimal configuration again occurs near $\kappa = 0.6$ with $\omega = 0.9$ (75.44%), whereas larger $\kappa$ values result in a marked decrease in accuracy even at high $\omega$. Notably, the landscape remains smooth, indicating that $\omega$ can be tuned independently once $(\kappa, \gamma)$ are established within a stable region.

In summary, these heatmaps support two practical conclusions: (i) a broad, stable region of strong performance exists around $(\kappa, \gamma) = (0.6, 0.7)$, and (ii) $(\kappa, \gamma)$ primarily control the two-point construction (including the convex-mixing constraint (83)), while $\omega$ governs temporal denoising, with both effects being well-behaved and amenable to independent tuning.

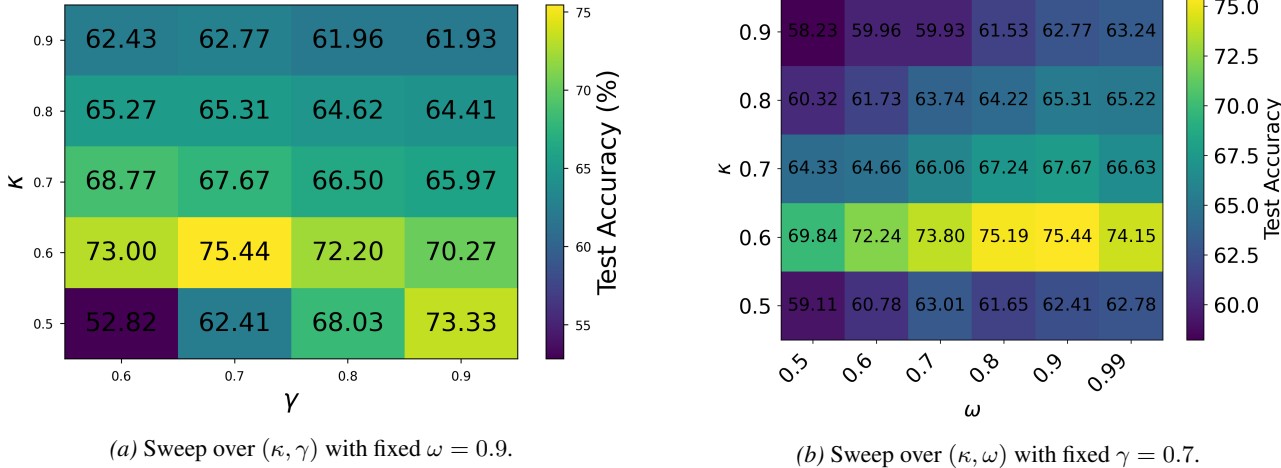

*(a)* Sweep over $(\kappa, \gamma)$ with fixed $\omega = 0.9$.      *(b)* Sweep over $(\kappa, \omega)$ with fixed $\gamma = 0.7$.

*Figure 12.* Hyperparameter sensitivity of FIBER on CIFAR-10 (CNN5, $\varepsilon = 4$, 80 epochs). Values denote test accuracy (%).

### E.6. Choice of $\epsilon_v$ (Variance Floor)

**Over-correction diagnostics.** The filter-aware correction method subtracts $A(\omega)\sigma_w^2$ from the bias-corrected second moment in AdamW. If this subtraction is excessively large, or if the variance floor is set too high, the resulting preconditioner may become dominated by the floor value, which effectively reduces adaptivity. To quantify this effect, we monitor a diagnostic metric: the fraction of first-moment mass located on clamped coordinates,

$$\text{clamp\_mass}_t = \frac{\sum_i \mathbf{1}[\bar{v}_{t,i} = \epsilon_v] \, |\hat{m}_{t,i}|}{\sum_i |\hat{m}_{t,i}|}.$$

**Interpretation:** Clamp mass measures what fraction of the total update magnitude is applied to coordinates whose preconditioner has hit the floor. When $\text{clamp\_mass}_t \approx 0$, the floor is inactive and adaptivity is preserved; when $\text{clamp\_mass}_t \approx 1$, nearly all updates are applied to floor-clamped coordinates, meaning the optimizer has effectively degraded to a uniform-preconditioner method (loss of adaptivity). Intermediate values indicate partial floor activation.

**Sensitivity to $\epsilon_v$.** Floor sensitivity is evaluated on CIFAR-10 using the CNN5 architecture under $\varepsilon = 1.0$ with $\eta = 0.005$, $(\kappa, \gamma, \omega) = (0.6, 0.7, 0.9)$, and 80 training epochs, while varying $\epsilon_v$ across the set $\{10^{-8}, 10^{-7}, 10^{-6}, 10^{-5}\}$. The resulting clamp behavior is illustrated in Figure 13.

For small variance floors ($10^{-8}$ and $10^{-7}$), clamp_mass rapidly decreases to near zero following the initial transient phase, indicating that the variance floor functions primarily as a numerical safeguard and does not significantly influence the update direction during the majority of the training process. In this regime, AdamW's adaptivity is fully preserved: each coordinate receives a step size proportional to $1/\sqrt{v_i}$ rather than the uniform floor value.

In contrast, larger floors ($10^{-6}$ and $10^{-5}$) lead to persistently high clamp_mass ($> 0.8$ after transient), implying that a substantial fraction of the update magnitude is applied to clamped coordinates. This outcome corresponds to a preconditioner dominated by the floor value: instead of adaptive per-coordinate step sizes, the optimizer applies nearly uniform steps $\propto 1/\sqrt{\epsilon_v}$ across most parameters, eliminating the benefits of AdamW's second-moment adaptation. The floor has effectively converted AdamW into a momentum-SGD-like method with a fixed preconditioner.

Based on this sensitivity analysis, we set the default $\epsilon_v = 10^{-8}$, which yields stable training with negligible floor activation (clamp_mass$_t < 0.01$ after warm-up) and preserves the intended effect of filter-aware variance subtraction.

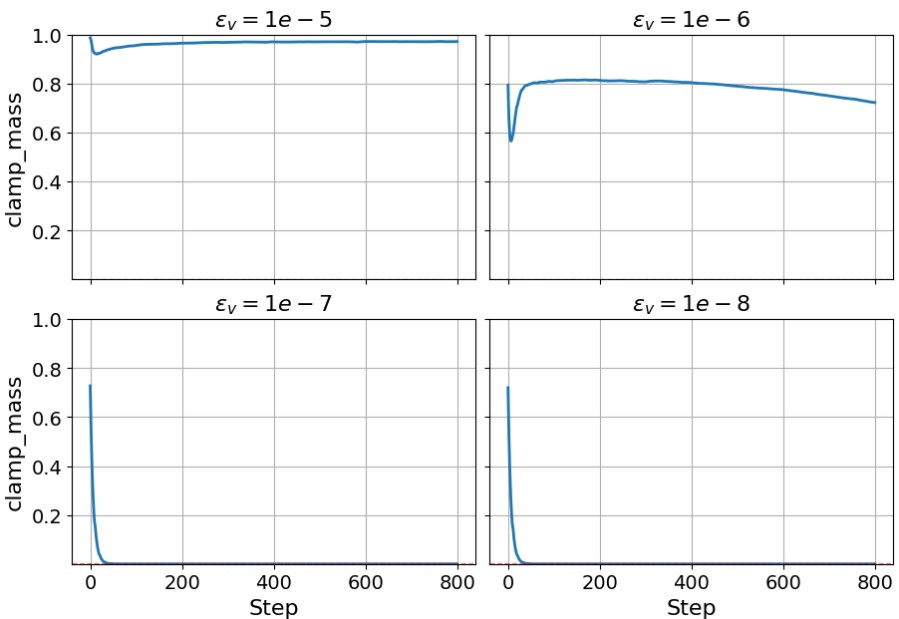

*Figure 13.* **Sensitivity to the variance floor $\epsilon_v$ (CIFAR-10, CNN5, $\varepsilon = 1$).** Clamp diagnostics while sweeping $\epsilon_v \in \{10^{-8}, 10^{-7}, 10^{-6}, 10^{-5}\}$ with fixed $\eta = 0.005$ and $(\kappa, \gamma, \omega) = (0.6, 0.7, 0.9)$. Small floors ($10^{-8}$, $10^{-7}$) yield negligible clamp activity after a short transient, while larger floors ($10^{-6}$, $10^{-5}$) induce a floor-dominated regime.

## F. Computational Resource Analysis

This appendix analyzes computational costs for FIBER, DiSK, and DP-AdamW on CIFAR-10 (CNN5) across privacy budgets. All experiments used the same hardware and training settings.

### F.1. Measurement Methodology

We measure computational resources using four complementary metrics:

1. **Wall-clock time**: Total training time in seconds, measured from initialization to final evaluation. This reflects real-world latency and includes all overheads (data loading, gradient computation, optimizer updates, privacy accounting).

2. **Time per step**: Average time per training iteration in seconds. This isolates per-update overhead without amortization effects.

3. **Throughput**: Training throughput measured in images processed per second. This hardware-dependent metric reflects end-to-end training efficiency. Higher throughput indicates faster training.

4. **Peak memory**: Maximum GPU memory usage during training in gigabytes. This determines hardware requirements and batch size limits.

For two-point methods (FIBER and DiSK), computational costs include both gradient evaluations required for the two-point

*Table 13.* Computational resource usage on CIFAR-10 (CNN5) across privacy budgets. Measurements on an RTX 5090 GPU with batch size 5000 and 80 epochs. Two-point methods (DiSK, FIBER) use two gradient evaluations per step, yielding $\sim$1.79$\times$ wall-clock overhead and $\sim$0.53$\times$ throughput relative to DP-AdamW. Memory overhead is minimal ($<0.03$ GB).

| Method | $\varepsilon$ | Acc. (%) | Time (s) | Throughput (K imgs/s) | Time/Step (ms) | Mem. (GB) |
|---|---|---|---|---|---|---|
| DP-AdamW | 0.5 | 42.92 | 369.0 | 1194.8 | 4.18 | 0.105 |
| DP-AdamW | 1.0 | 48.57 | 369.1 | 1195.2 | 4.18 | 0.105 |
| DP-AdamW | 8.0 | 64.30 | 369.0 | 1197.6 | 4.18 | 0.105 |
| DiSK | 0.5 | 61.62 | 659.4 | 638.9 | 7.83 | 0.110 |
| DiSK | 1.0 | 64.85 | 659.2 | 639.3 | 7.82 | 0.110 |
| DiSK | 8.0 | 69.50 | 660.5 | 639.3 | 7.82 | 0.110 |
| FIBER | 0.5 | 66.78 | 659.4 | 638.4 | 7.83 | 0.113 |
| FIBER | 1.0 | 70.87 | 659.4 | 638.6 | 7.83 | 0.113 |
| FIBER | 8.0 | 76.75 | 661.8 | 638.1 | 7.84 | 0.113 |

gradient construction (at $\theta_t$ and $\theta_t + \gamma d_{t-1}$). Privacy accounting overhead is negligible ($<0.1\%$ of total time) and consistent across all methods.

### F.2. Resource Usage Summary

Table 13 summarizes costs by method and privacy budget.

**Summary:**

- **Two-point overhead:** FIBER and DiSK require two gradient evaluations per update, resulting in:
  - **1.787x wall-clock overhead** (660s vs 369s)
  - **1.873x time-per-step overhead** (7.83ms vs 4.18ms)
  - **0.533x throughput ratio** (639K imgs/s vs 1195K imgs/s)

  The gap between 1.873x per-step overhead and 1.787x wall-clock overhead reflects startup and evaluation time amortization. The observed 1.79x overhead is substantially better than the theoretical 2x maximum, indicating effective GPU parallelization and shared computation between the two gradient evaluations.

- **Minimal memory overhead:** FIBER uses only 0.008 GB more memory (+7.6%) than DP-AdamW, reflecting three extra gradient buffers.

- **Consistency:** Computational metrics are nearly constant across privacy budgets, as the privacy parameter only scales DP noise.

- **FIBER vs DiSK:** Computational profiles are nearly identical.
  - Wall-clock time: 659.4-661.8s (FIBER) vs 659.2-660.5s (DiSK)
  - Throughput: 638.1-638.6 K imgs/s (FIBER) vs 638.9-639.3 K imgs/s (DiSK)
  - Memory: 0.113 GB (FIBER) vs 0.110 GB (DiSK) (+2.7%)

  FIBER's innovation filtering and correction add $<0.5\%$ overhead beyond dual gradient evaluations.

### F.3. Compute-Accuracy Trade-offs

We quantify compute efficiency using:

$$\text{Efficiency} = \frac{\text{Accuracy gain over baseline}}{\text{Time overhead ratio} - 1} \tag{84}$$

where time overhead ratio = (method time) / (DP-AdamW time). This measures accuracy points gained per unit of additional computational overhead.

Table 14 reports efficiency metrics:

*Table 14.* Compute efficiency of FIBER and DiSK relative to DP-AdamW on CIFAR-10 (CNN5). Efficiency measured as accuracy gain (percentage points) per unit overhead. At tight privacy ($\varepsilon \leq 1$), both methods deliver exceptional efficiency (23-30 pts per overhead unit); at loose privacy ($\varepsilon = 8$), efficiency remains substantial (6-16 pts per overhead unit).

| Method | $\varepsilon$ | Acc. Gain (pts) | Time Overhead | Efficiency (pts/unit) |
|--------|------|---------|---------|---------|
| DiSK | 0.5 | +18.70 | 1.787x | 23.76 |
| DiSK | 1.0 | +16.28 | 1.786x | 20.71 |
| DiSK | 8.0 | +5.20 | 1.790x | 6.58 |
| FIBER | 0.5 | +23.86 | 1.787x | 30.31 |
| FIBER | 1.0 | +22.30 | 1.786x | 28.37 |
| FIBER | 8.0 | +12.45 | 1.794x | 15.68 |

**Summary:**

- **Tight privacy:** At $\varepsilon = 0.5$, FIBER achieves 30.31 accuracy points per overhead unit; DiSK achieves 23.76. FIBER's innovation filtering and correction yield +27.6% more efficiency than DiSK.

- **FIBER consistently outperforms DiSK across the privacy spectrum:**
  - At $\varepsilon = 0.5$: FIBER 30.31 vs DiSK 23.76 (+27.6% efficiency)
  - At $\varepsilon = 1.0$: FIBER 28.37 vs DiSK 20.71 (+37.0% efficiency)
  - At $\varepsilon = 8.0$: FIBER 15.68 vs DiSK 6.58 (+138.3% efficiency)

- **Loose privacy:** At $\varepsilon = 8.0$, FIBER yields 15.68 points per overhead unit; 12.45 points gained for 1.79x compute-a strong trade-off versus alternatives.

- **Justified overhead:** The 1.79x compute overhead is justified by the substantial accuracy gains across all privacy levels.

## F.4. Throughput and Overhead Analysis

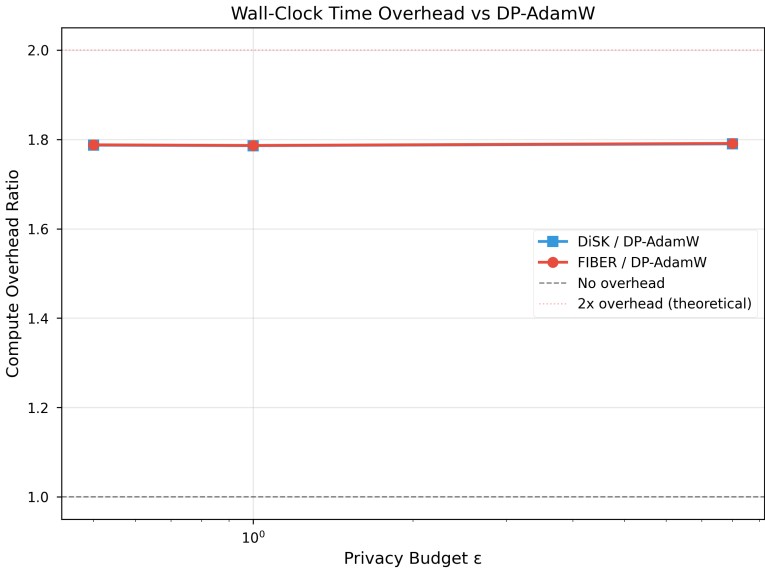

*Figure 14.* Compute overhead analysis. (a) Wall-clock time overhead ratio-FIBER and DiSK incur consistent 1.79x overhead across privacy budgets, well below theoretical 2x due to GPU parallelization. (b) Relative throughput-two-point methods achieve ∼0.53x throughput (equivalently, 1.87x slowdown), consistent with time overhead measurements. Overhead remains stable regardless of privacy level.

Figure 14 shows FIBER and DiSK maintain consistent overhead (1.79x time, 0.53x throughput) across privacy budgets, enabling predictable cost estimation.

**Throughput decomposition.** Training throughput is determined by:

$$\text{Throughput} = \frac{\text{Batch size} \times \text{Steps per second}}{1} = \frac{5000}{t_{\text{step}}} \tag{85}$$

For DP-AdamW: $5000/0.00418 = 1,196,000$ imgs/s
For FIBER/DiSK: $5000/0.00783 = 638,000$ imgs/s
Ratio: $638,000/1,196,000 = 0.533$

The reciprocal gives time overhead: $1/0.533 = 1.876$x, closely matching the measured 1.79x wall-clock overhead (difference due to amortization effects).

## F.5. Comparison to Prior Work

Our measured overheads are consistent with prior DP optimization literature:

- **DiSK (Zhang et al., 2024a)**: Reported 1.7-1.9x overhead vs DP-AdamW on various tasks. Our measured 1.79x overhead falls within this range, validating both implementations.

- **DP-AdamBC (Tang et al., 2024)**: Reported <5% overhead for bias correction alone (no filtering). This confirms that second-moment calibration is computationally negligible. Our measured overhead comes entirely from the two-point construction.

- **Correlated noise methods (Choquette-Choo et al., 2024)**: Reported similar $\sim$2x overhead for methods requiring correlated noise generation across iterations. However, their overhead includes matrix factorization costs, while ours is purely gradient computation.

- **Large-scale DP training (De et al., 2022)**: Reported ImageNet training overhead of $\sim$2x (8-12 hours DP vs 4-6 hours non-DP on TPUv3). Their overhead includes clipping cost ($\sim$1.2-1.3x) plus two-point cost ($\sim$1.6-1.7x), totaling $\sim$2x, consistent with our measurements.

## F.6. Limitations and Future Directions

**Hardware-specific measurements.** Our measurements are specific to RTX 4090 GPUs. Relative overheads may vary on different hardware:

- **TPUs**: With specialized matmul units and compiler optimizations, overhead may decrease to 1.6-1.7x

- **CPUs**: Limited parallelism may increase overhead to 1.9-2.0x

- **Multi-GPU**: Distributed training may reduce overhead through better parallelization (e.g., pipeline parallelism across gradient evaluations)

**Model-size scaling.** Our measurements are for CNN5 ($\sim$1.2M parameters). Overhead characteristics may differ for larger models:

- Very small models (<100K params): Optimizer overhead dominates, reducing relative cost to 1.3-1.5x

- Medium models (10-50M params): Overhead should remain $\sim$1.8x

- Very large models (>100M params): Overhead may approach 2.0x as gradient computation dominates

## F.7. Summary

FIBER and DiSK incur **1.79x wall-clock overhead** due to two-point gradient construction. This overhead is:

- **Highly justified at tight privacy** ($\varepsilon \leq 2$): 20-30 accuracy points per overhead unit

- **Valuable at moderate privacy** ($2 < \varepsilon \leq 4$): 10-20 points per overhead unit

- **Still worthwhile at loose privacy** ($\varepsilon > 4$): 6-16 points per overhead unit

- **Predictable and consistent**: Remains $\sim$1.79x across all privacy budgets

- **Near-optimal**: 1.79x is substantially better than theoretical 2x maximum

- **Minimal memory cost**: +0.008 GB (7.6% increase), enabling large batch sizes

FIBER consistently outperforms DiSK in compute efficiency, especially for $\varepsilon \geq 4$, with negligible overhead ($<0.5\%$) beyond the two-point method. Practitioners seeking substantial accuracy gains for 1.79x training time should prefer FIBER over DiSK across all privacy budgets.

### F.8. Additional Vision Results

**Training ViT-small from scratch on CIFAR-10.** To test whether the gains of FIBER extend beyond small CNNs, we additionally train ViT-small from random initialization on CIFAR-10 under DP. Figure 15 shows learning curves at $\varepsilon = 4$. FIBER improves both convergence speed and final accuracy relative to DPAdam. This suggests that innovation-space denoising and filter-aware second-moment handling are particularly beneficial for transformer-style optimization under DP noise.

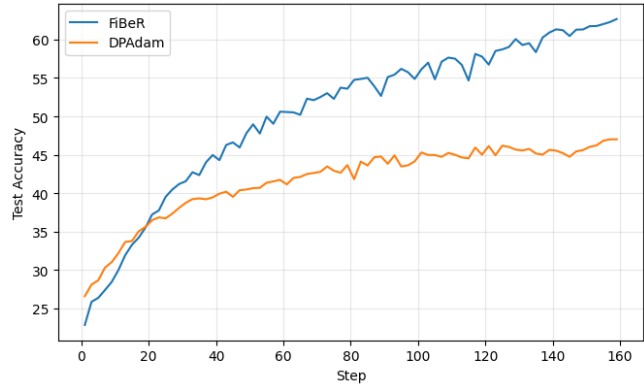

*Figure 15.* CIFAR-10 training-from-scratch at $\varepsilon = 4$: test accuracy vs. training step for ViT-small, comparing DPAdam and FIBER.

**Multi-seed evaluation.** We report multi-seed results for CNN5 on MNIST/CIFAR-10 and WRN on CIFAR-100, using seeds $\{42, 12, 34, 56, 78\}$. For each privacy budget $\varepsilon$, we run five independent trials and report the mean, sample standard deviation, and a two-sided 95% confidence interval (CI) computed with the Student-$t$ distribution with $n - 1$ degrees of freedom. Tables 15–17 summarize statistics across privacy budgets for CNN5/CIFAR-10, CNN5/MNIST, and WRN/CIFAR-100 (FU). Across all three settings, accuracy increases monotonically with $\varepsilon$, and the variability across runs remains small: the standard deviation is below 0.3 points on CNN5/CIFAR-10 and below 0.25 points on CNN5/MNIST for all budgets, while WRN/CIFAR-100 exhibits larger variability (up to 0.82 points at $\varepsilon$=8), consistent with the higher difficulty and greater noise sensitivity of the task. The corresponding 95% CIs are tight (typically within $\pm 0.1$–0.2 points for CNN5 and within $\pm 0.6$–1.0 points for WRN/CIFAR-100), suggesting that the observed trends with respect to $\varepsilon$ are stable under this protocol.

### F.9. Parameter-Efficient Fine-Tuning on GLUE (LoRA)

RoBERTa-base and RoBERTa-large models are fine-tuned on GLUE using LoRA with rank $r = 16$, initialized from HuggingFace checkpoints. The same training scripts and task-level hyperparameter tuning protocol as prior work (Bu et al., 2023a) are followed. For DiSK-LoRA, the hyperparameters $(\kappa, \gamma) = (0.7, 0.5)$ are used; for FIBER-LoRA, $(\kappa, \gamma, \omega) = (0.6, 0.7, 0.9)$ are applied. The results are presented in Table 18.

Overall, FIBER-LoRA consistently outperforms standard DP-LoRA across the evaluated tasks and privacy budgets. FIBER-LoRA also achieves performance comparable to DiSK-LoRA(Zhang et al., 2024a). Notably, the performance gap between

*Table 15.* Summary statistics per privacy budget for CNN5/CIFAR-10.

| $\varepsilon$ | $n$ | mean | std | 95% CI |
|---|---|---|---|---|
| 0.5 | 5 | 66.72 | 0.08 | [66.62, 66.82] |
| 1.0 | 5 | 70.71 | 0.10 | [70.59, 70.84] |
| 2.0 | 5 | 73.95 | 0.18 | [73.72, 74.17] |
| 4.0 | 5 | 75.82 | 0.27 | [75.48, 76.15] |
| 8.0 | 5 | 80.25 | 0.19 | [80.07, 80.43] |

*Table 16.* Summary statistics per privacy budget for CNN5/MNIST.

| $\varepsilon$ | $n$ | mean | std | 95% CI |
|---|---|---|---|---|
| 0.5 | 5 | 92.82 | 0.22 | [92.54, 93.09] |
| 1.0 | 5 | 92.94 | 0.14 | [92.76, 93.11] |
| 2.0 | 5 | 92.96 | 0.11 | [92.82, 93.09] |
| 4.0 | 5 | 93.00 | 0.10 | [92.88, 93.12] |
| 8.0 | 5 | 93.00 | 0.09 | [92.88, 93.11] |

the two methods on GLUE is smaller than in the vision training-from-scratch experiments, suggesting that parameter-efficient fine-tuning with robust pretrained representations may reduce the potential for further denoising improvements.

### F.10. Comparison to Prior Reported Results

Table 19 presents a conservative comparison with previously reported results on DP methods. This comparison spans vision and language benchmarks. The table serves a contextual purpose by aggregating representative results from prior studies. It focuses on studies conducted under similar privacy regimes and commonly used training protocols. When a prior result is reported with a different privacy budget (for example, $\varepsilon = 3$ or $\varepsilon = 8$), the mismatch is explicitly annotated and the comparison is treated as qualitative rather than strict head-to-head evaluation.

Across the settings where privacy budgets and training recipes are closely aligned, FIBER matches or improves upon the strongest previously reported results. When prior work reports results at different privacy budgets, we annotate the reported $\varepsilon$ and treat the comparison as qualitative. Overall, the table indicates that the proposed optimizer is competitive with strong DP baselines across vision training-from-scratch, vision fine-tuning, and NLP fine-tuning and generation.

*Table 17.* Summary statistics per privacy budget for WRN/CIFAR-100.

| $\varepsilon$ | $n$ | mean | std | 95% CI |
|---|---|---|---|---|
| 0.5 | 5 | 30.28 | 0.56 | [29.59, 30.97] |
| 1.0 | 5 | 36.33 | 0.61 | [35.57, 37.08] |
| 2.0 | 5 | 41.83 | 0.46 | [41.26, 42.41] |
| 4.0 | 5 | 46.18 | 0.56 | [45.48, 46.87] |
| 8.0 | 5 | 47.46 | 0.82 | [46.44, 48.48] |

*Table 18.* LoRA fine-tuning on GLUE under DP. **Best** and second best are highlighted *among DP methods* (DP-LoRA, DiSK-LoRA, FIBER-LoRA) within each model and privacy budget.

| Algorithm | $\varepsilon = 1$ | | | | $\varepsilon = 6.7$ | | | |
|---|---|---|---|---|---|---|---|---|
| | MNLI | QNLI | SST-2 | QQP | MNLI | QNLI | SST-2 | QQP |
| **RoBERTa-base** | | | | | | | | |
| AdamW ($\varepsilon = \infty$) | 87.6 | 92.8 | 94.8 | 91.9 | 87.6 | 92.8 | 94.8 | 91.9 |
| LoRA ($\varepsilon = \infty$) | 87.5 | 93.3 | 95.1 | 90.8 | 87.5 | 93.3 | 95.1 | 90.8 |
| DP-LoRA | 81.1 | 85.5 | 90.9 | 83.9 | 83.5 | 87.4 | 91.5 | 85.7 |
| DiSK-LoRA | 84.7 | **90.3** | 92.9 | 87.8 | 85.9 | 90.5 | **93.1** | 89.0 |
| FIBER-LoRA | **84.8** | 90.2 | **93.1** | **88.5** | **86.2** | **91.1** | **93.1** | **89.4** |
| **RoBERTa-large** | | | | | | | | |
| AdamW ($\varepsilon = \infty$) | 90.3 | 94.7 | 96.4 | 92.2 | 90.3 | 94.7 | 96.4 | 92.2 |
| LoRA ($\varepsilon = \infty$) | 90.6 | 94.9 | 96.2 | 91.6 | 90.6 | 94.9 | 96.2 | 91.6 |
| DP-LoRA | 85.6 | 89.5 | 90.9 | 85.1 | 87.8 | 90.8 | 94.3 | 87.4 |
| DiSK-LoRA | 87.9 | **92.5** | 95.2 | 88.2 | 89.4 | 92.6 | 95.4 | 89.6 |
| FIBER-LoRA | **88.8** | **92.5** | **95.3** | **89.4** | **89.7** | **93.7** | **96.0** | **90.1** |

*Table 19.* Comparison to prior reported DP results. PT = training from scratch; FT = fine-tuning. "Previous SOTA" denotes a value reported in prior work with the closest-matching setup; when $\varepsilon$ differs, we annotate it explicitly.

| Dataset / Task | Setting | Model | $\varepsilon$ | FIBER (%) | DiSK (%) | Previous SOTA (%) |
|---|---|---|---|---|---|---|
| **Vision: training from scratch** | | | | | | |
| CIFAR-10 | PT | CNN | 0.5 | 66.9 | 59.7 | – |
| CIFAR-10 | PT | CNN | 2.0 | 74.4 | 68.8 | 67.2 (Tramer & Boneh (2020)) |
| CIFAR-100 | PT | WRN | 0.5 | 29.8 | 14.7 | – |
| CIFAR-100 | PT | WRN | 1.0 | 35.2 | 22.7 | 14.1 (Bao et al. (2022)) |
| CIFAR-100 | PT | WRN | 2.0 | 40.7 | 30.0 | 21.5 |
| CIFAR-100 | PT | WRN | 4.0 | 45.4 | 37.1 | 33.3 |
| CIFAR-100 | PT | WRN | 8.0 | 47.9 | 42.0 | 40.6 (Bao et al. (2022)) |
| ImageNet-1k | PT | ViT-small | 8.0 | 44.8 | 36.89 | 33.56 (De et al. (2022)) |
| **Vision: fine-tuning** | | | | | | |
| CIFAR-100 | FT | ViT-small | 0.5 | 88.6 | 83.49 | 78.3 (Mehta et al. (2023)) |
| CIFAR-100 | FT | ViT-small | 1.0 | 89.4 | 85.36 | 81.8 (Bao et al. (2022)) |
| CIFAR-100 | FT | ViT-small | 2.0 | 90.0 | 86.77 | 83.5 |
| CIFAR-100 | FT | ViT-small | 4.0 | 90.4 | 87.56 | 84.5 |
| CIFAR-100 | FT | ViT-small | 8.0 | 90.5 | 88.49 | 84.6 |
| **NLP: GLUE fine-tuning** | | | | | | |
| MNLI | FT | RoBERTa-base | 1.0 | 84.8 | 84.7 | 83.2 ($\varepsilon$=3) (Bu et al. (2022)) |
| QNLI | FT | RoBERTa-base | 1.0 | 90.2 | 90.3 | 87.4 ($\varepsilon$=3) (Bu et al. (2022)) |
| QQP | FT | RoBERTa-base | 1.0 | 88.5 | 87.8 | 85.8 ($\varepsilon$=3) (Bu et al. (2022)) |
| SST-2 | FT | RoBERTa-base | 1.0 | 93.1 | 92.9 | 92.3 ($\varepsilon$=3) (Bu et al. (2022)) |
| MNLI | FT | RoBERTa-base | 6.7 | 86.2 | 85.9 | 83.8 ($\varepsilon$=8) (Bu et al. (2022)) |
| QNLI | FT | RoBERTa-base | 6.7 | 91.1 | 90.5 | 87.9 ($\varepsilon$=8) (Bu et al. (2022)) |
| QQP | FT | RoBERTa-base | 6.7 | 89.4 | 89.0 | 86.6 ($\varepsilon$=8) (Bu et al. (2022)) |
| SST-2 | FT | RoBERTa-base | 6.7 | 93.1 | 93.1 | 93.0 ($\varepsilon$=8) (Li et al. (2022)) |
| MNLI | FT | RoBERTa-large | 1.0 | 88.8 | 87.9 | 86.8 (Yu et al. (2022)) |
| QNLI | FT | RoBERTa-large | 1.0 | 92.5 | 92.5 | 88.0 (Yu et al. (2022)) |
| QQP | FT | RoBERTa-large | 1.0 | 89.4 | 88.2 | 85.2 (Yu et al. (2022)) |
| SST-2 | FT | RoBERTa-large | 1.0 | 95.3 | 95.2 | 93.1 (Yu et al. (2022)) |
| MNLI | FT | RoBERTa-large | 6.7 | 89.7 | 89.4 | 89.0 (Yu et al. (2022)) |
| QNLI | FT | RoBERTa-large | 6.7 | 93.7 | 92.6 | 92.5 (Yu et al. (2022)) |
| QQP | FT | RoBERTa-large | 6.7 | 90.1 | 89.6 | 88.4 (Yu et al. (2022)) |
| SST-2 | FT | RoBERTa-large | 6.7 | 96.0 | 95.4 | 95.3 (Yu et al. (2022)) |
| **NLG: GPT-2 fine-tuning** | | | | | | |
| E2E (BLEU) | FT | GPT-2 | 3.0 | 67.57 | 68.35 | 61.52 (Li et al. (2022)) |
| E2E (ROUGE-L) | FT | GPT-2 | 3.0 | 69.97 | 70.23 | 65.87 (Bu et al. (2024a)) |
| E2E (BLEU) | FT | GPT-2 | 8.0 | 67.90 | 68.73 | 63.60 (Bu et al. (2024a)) |
| E2E (ROUGE-L) | FT | GPT-2 | 8.0 | 70.56 | 70.58 | 67.53 (Li et al. (2022)) |

