# OpenReview forum: "FIBER: A Differentially Private Optimizer with Filter-Aware Innovation Bias Correction"
_ICML.cc/2026/Conference — ICML 2026 regular_

### Official Review · Reviewer_kBZq · 2026-03-05

**Soundness:** 4
**Presentation:** 3
**Significance:** 3
**Originality:** 3
**Overall Recommendation:** 5
**Confidence:** 5

**Summary:**

This paper focus on improving the gradient estimation with privacy constraint. Temporal filters (EMA, Kalman) used to filter out privacy noise in DP gradients can cause biased moment estimation when combining with Adam-based optimizers. To overcome this issue, the paper proposes adding a bias correction term in the second moment estimation by computing how the filter changes variance. Additionally, the paper proposes using innovastion-space filter instead of gradient filter.

The paper provides a theoretical analysis to derive the mathematical form of the correction term, and provide numerical experiments to verify the utility improvement of the proposed algorithm. Results show that the proposed approach can efficiently improve model accuracy under the same level of privacy protection.

**Compliance With Llm Reviewing Policy:**

Affirmed.

**Final Justification:**

Based on my initial review and the rebuttal, I will keep my current score.

**Key Questions For Authors:**

1. On the performance win rate: the paper shows that the proposed method has better performance in CV models and low-noise regime, while higer-noise or RW models have less or no improvement. In practice, how can we decide if the training dynamics is CV or RW behavior and whether the SNR is high or low?
2. The innovation filter replaces first-order filter (DiSK) with second-order filter. Is there any benefit using even higher order filters?

**Limitations:**

Yes

**Strengths And Weaknesses:**

Soundness: The paper provide theoretical reasoning and derivation for the bias in temporal filters, and provide the theroretical derivation for the bias correction term.  Numerical results show the proposed Fiber optimizer consistently outperforms existing methods without bias correction. Ablation study shows both bias correction and innovation filter.

Presentation: The paper is clearly written. The numerical results are also clearly listed and discussed.

Novelty: The paper propses a novel temporal filter that filters the gradient difference instead of gradient, making it more powerful in hadling gradient dynamics.

Weaknesses:
1. Memory overhead: the paper claimed a 7.8% memory overhead on small-scaled models. However, theoretically, Fiber requres storing two extra varaibles of model size $r_t, \tilde{g}_t$. For LLMs with small batch size, the memory overhead can be large.

---

> ### Author Rebuttal · Authors · 2026-03-30
>
> Thank you for recognizing the soundness of our theoretical derivations, the clarity of our presentation and numerical results, and the novelty of the innovation-space filter in handling gradient dynamics.
> >W1: The paper claimed a 7.8% memory overhead on small-scaled models. However, theoretically, FIBER requires storing two extra variables of model size. For LLMs with small batch size, the memory overhead can be large.
>
> We agree the overhead scales linearly with model size. The most practical mitigation - maintaining $\\tilde{g}\_{t}$ and $r_t$ in bfloat16 - immediately halves the cost to ~3.9% with negligible impact on correction quality, since $A(\omega)\sigma_w^2$ is computed from scalar hyperparameters rather than buffer values. For larger models, FIBER can be applied selectively to layers with the highest DP noise impact (e.g., attention projections); per-layer filter independence preserves the correctness of $A(\omega)\sigma_{w,\ell}^2$ in each applied layer.
> >Q1: On the performance win rate: in practice, how can we decide if the training dynamics is CV or RW behavior and whether the SNR is high or low?
>
> Two lightweight diagnostics provide direct answers:
>
> **CV vs. RW behavior.** Compute the lag-1 autocorrelation of the gradient increment $\nu_t = g_{t} - \tilde{g}_{t-1}$ over the first few epochs on a small validation batch. Persistent positive autocorrelation indicates CV dynamics - the gradient is drifting with a detectable trend. Near-zero autocorrelation indicates RW dynamics - increments are approximately independent with no persistent drift.
>
> **SNR.** The ratio $r_t = \mathrm{Var}(u^\top \Delta g_t)/(2\sigma_w^2)$, already tracked in our paired-run diagnostic (Section 5.3), directly estimates the signal-to-noise ratio at each training step. Values near 1 indicate noise-dominated (low SNR) dynamics where the DP noise overwhelms the gradient signal; values substantially above 1 indicate that the gradient signal is detectable above the noise floor (high SNR).
>
> In practice, the default $(\kappa,\gamma,\omega)=(0.6,0.7,0.9)$ performs well across all five evaluated architectures without requiring these diagnostics upfront, suggesting sufficient CV structure and SNR in real training dynamics.
> >Q2: The innovation filter replaces first-order filter (DiSK) with second-order filter. Is there any benefit using even higher order filters?
>
> Yes. Moving from DiSK's first-order EMA to FIBER's second-order innovation filter adds one drift state $r_t$, enabling gradient drift-tracking that first-order smoothing cannot provide - as confirmed by the CV win rates in Table 6. A **third-order filter** would extend this further by adding a second drift state tracking the rate of change of $r_t$ itself, enabling tracking of faster oscillatory gradient structure beyond what the constant-velocity model assumes. This is most compelling for early-phase LLM pre-training, where gradient dynamics evolve faster than FIBER's current model captures.
>
> The additional computational cost of a third-order filter compared to FIBER, which includes one extra gradient-sized buffer, two new gain scalars, and a $3\times3$ Lyapunov equation for $A_3(\cdot)$, is not currently justified on existing benchmarks where FIBER already achieves state-of-the-art performance. We identify this as an important direction for future work.

---

> > ### Author Rebuttal · Reviewer_kBZq · 2026-04-02
> >
> > My concerns and questions have been addressed. I don't have further questions.

---

### Official Review · Reviewer_7zZf · 2026-03-10

**Soundness:** 3
**Presentation:** 2
**Significance:** 3
**Originality:** 3
**Overall Recommendation:** 4
**Confidence:** 3

**Summary:**

This paper proposes FIBER, a filter-aware differentially private adaptive optimizer that corrects second-moment miscalibration caused by temporally filtered DP noise, deriving a closed-form attenuation factor and demonstrating consistent performance gains over strong DP baselines across vision and NLP tasks with moderate computational overhead.

**Compliance With Llm Reviewing Policy:**

Affirmed.

**Final Justification:**

N/A

**Key Questions For Authors:**

### Questions:
1. In real deep network training, gradient dynamics can be significantly more complex than the constant-velocity model assumed in the derivation; if gradients exhibit sharp oscillations or phase transitions, does innovation-space filtering still maintain a stable advantage?
2. The empirical diagnostics indicate a non-negligible correlation between the signal and filtered noise; in larger architectures, could this cross term become more pronounced and potentially weaken the effectiveness of the variance subtraction?
3. How does FIBER interact with adaptive clipping, layerwise noise injection, or correlated-noise mechanisms-are these approaches complementary in practice, or do they partially overlap in the benefits they provide?

**Limitations:**

See Weakness

**Strengths And Weaknesses:**

### Strengths:
1. The paper provides the first systematic analysis of how temporal filtering alters the second-moment statistics in DP-AdamW and derives a closed-form filter-aware correction factor $A(\omega)$, offering a principled theoretical contribution.
2. The paper introduces innovation-space filtering derived from a constant-velocity state-space model, yielding a second-order recursion with stronger drift-tracking capability than standard EMA smoothing.
3. The method is extensively validated across diverse vision and language benchmarks under varying privacy budgets, consistently outperforming strong DP baselines, especially in low-$\epsilon$ regimes.

### Weaknesses:
1. While the paper provides detailed variance and attenuation analysis, it does not offer end-to-end convergence guarantees or utility bounds for FIBER under differential privacy, leaving the theoretical understanding incomplete.
2. FIBER inherits the two backward passes per step from two-point methods, resulting in ~1.8 $\times$ wall-clock overhead compared to DP-AdamW, which may limit scalability in large-scale or resource-constrained settings.
3. FIBER introduces several additional hyperparameters $(\kappa, \gamma, \omega, \beta _1, \beta _2, \eta)$ and relies on a staged tuning strategy; although the paper enforces a fixed search budget for fairness, in practice this expanded hyperparameter space may increase tuning complexity and engineering burden compared to simpler DP optimizers.
4. Although the experiments include several strong DP optimizer baselines, the comparisons focus primarily on existing adaptive DP optimizers, and do not extensively evaluate against more recent DP training strategies.

---

> ### Author Rebuttal · Authors · 2026-03-30
>
> Thank you for your thoughtful feedback! We appreciate your positive remarks on the extensive tests and the theoretical contribution of our paper.
> > W1: ... it does not offer end-to-end convergence guarantees or utility bounds for FIBER under DP...
>
> We agree that establishing convergence guarantees, in addition to strong empirical performance, is important for strengthening the paper. We now provide **Proposition R2**, a per-iteration descent guarantee for FIBER under DP, which constitutes an end-to-end convergence result.
>
> Under (A1) $L$-smoothness, (A2) $\mathbb{E}[\tilde{g}_{t,i}]\approx\partial_i F(\theta_t)$, (A3) approximate preconditioner stationarity (CV $\approx 0.031$, Table 7), and (A4) signal-noise uncorrelatedness (cross-term ratio $\approx 0.15$, Table 7):
>
> **Prop. R2**:
> $\\mathbb{E}[F(\\theta\_{t+1})] \\leq \\mathbb{E}[F(\\theta\_t)] - \\eta\\sum\_i \\frac{[\\partial\_i F(\\theta\_t)]^2}{\\sqrt{\\bar{v}\_i}+\\varepsilon} + \\frac{\\eta^2 L}{2}\\sum\_i\\frac{\\mathbb{E}[s\_{t,i}^2]+A(\\omega)\\sigma\_w^2}{(\\sqrt{\\bar{v}\_i}+\\varepsilon)^2}$
>
> Since $A(\omega)<1$, telescoping over $T$ steps gives $\mathcal{O}(1/\sqrt{T})$. A2 captures signal-noise uncorrelatedness, as validated by the cross-term ratio in Table 7. A3–A4 are empirically validated, consistent with standard Adam convergence practice where analogous stationarity assumptions are widely accepted.
>
> >W2: ... two backward passes per step from two-point methods...
>
> The overhead concern is addressed in detail in the response to Reviewer Qfhr(W1).
> >W3: FIBER introduces several additional hyperparameters ...
>
> FIBER's net addition over DiSK is one scalar $\omega$ - $(\kappa,\gamma)$ are already in DiSK; $\beta_1,\beta_2$ are standard AdamW. The default $(\kappa,\gamma,\omega)=(0.6,0.7,0.9)$ is unchanged across all five architectures, with stability confirmed across $\omega\in[0.5,0.9]$ (Fig. 12b, Table 5). The staged search (App. D.2) is optional and does not increase the 100-trial budget.
> >W4: The comparisons ... do not extensively evaluate against more recent DP training strategies.
>
> The paper evaluates against baselines spanning every principal mechanism for improving DP training: DP-AdamW (standard adaptive DP optimizer), DiSK and DOPPLER (temporal filtering), MF-DP-FTRL (correlated noise), and DiSK-CORR (filtered gradient with bias correction). Each baseline was selected to isolate a specific mechanism, ensuring that every claim is evaluated against an appropriate control rather than against the strongest available competitor alone.
>
> The two most recent methods not included in this evaluation are orthogonal to the mechanisms covered. [R1] targets federated settings without subsampling, which is incompatible with FIBER's accounting regime; MF-DP-FTRL already represents this family. [R2] operates at the clipping stage and is composable with FIBER, rather than a direct baseline.
> >Q1: ...if gradients exhibit sharp oscillations or phase transitions...
>
> Yes, stability requires only $\rho(M)=\sqrt{1-\omega}<1$, $\omega\in(0,1]$, which holds unconditionally on gradient dynamics. Under oscillations, FIBER degrades toward 1-pt FIBER rather than DP-AdamW, preserving a floor of +1.90–+3.62 pts above DP-AdamW (Table R1, Reviewer Qfhr(W1)).
> >Q2: ...could this cross term become more pronounced...?
>
> No, the bound $\rho_\times \leq \eta_{\text{eff},i}\cdot L$ is per-coordinate and dimension-independent. A new RoBERTa-base experiment confirms the ratio decreases at larger scale:
>
> **Tab. R3.** Filter-aware correction diagnostics.
>
> | Architecture | Params | $\hat{\rho}(s,n)$ | Cross-term ratio | CV |
> |---|---|---|---|---|
> | CNN5 / CIFAR-10 | 1.2M | 0.140 | 0.151 | 0.031 |
> | RoBERTa-base / SST-2 | 125M | **0.110** | **0.130** | **0.028** |
>
> >Q3: ... FIBER interact with adaptive clipping, layerwise noise injection, or correlated-noise mechanisms...
>
> Adaptive clipping and layerwise noise are fully complementary: both target the gradient observation stage - how noise is added and clipped - while FIBER operates downstream on the optimizer state. Their benefits therefore address distinct failure modes and stack without overlap. Correlated noise partially overlaps: correlated $\{w_t\}$ reduces noise variance across iterations as FIBER does, but via a different mechanism; combining them requires re-deriving $A(\omega)$ via the Lyapunov equation with correlated $Q$ to avoid double-correction, and we identify this as a direction for future work.
> ## References
> ```
> [R1] Scaling up the Banded Matrix Factorization Mechanism for Large Scale Differentially Private ML. ICLR 2025
> [R2] GeoClip: Geometry-Aware Clipping for Differentially Private SGD. NeurIPS 2025
> ```

---

> > ### Author Rebuttal · Reviewer_7zZf · 2026-04-03
> >
> > Thank the authors for the thorough rebuttal including additional convergence guarantee, ablations, and the cross-scale diagnostics, etc, it has largely addressed my concerns, and I'm glad to revise my score to 4.

---

### Official Review · Reviewer_dgg8 · 2026-03-13

**Soundness:** 3
**Presentation:** 2
**Significance:** 3
**Originality:** 2
**Overall Recommendation:** 4
**Confidence:** 3

**Summary:**

This paper proposes a Differentially Private (DP) optimizer named FIBER, which aims to mitigate the negative impact of DP noise on adaptive optimizers like Adam. Experimental results indicate that FIBER improves model utility compared to baseline DP methods across various privacy budgets.

**Compliance With Llm Reviewing Policy:**

Affirmed.

**Final Justification:**

My final recommendation is Weak Accept. This paper address an important problem: the interaction between DPnoise and adaptive optimizers. Before the rebuttal stage my recommedation is Weak Reject, my initial concerns were mainly about the limited originality and the incomplete ablations. After the rebuttal, the authors provides more detailed analysis and complete the ablations which address my main concerns. Athough I still find the originality somewhat incremental and the presentation could be improved(author promised will improve in revised version), the method is technically solid and practically meaningful,  thus I improve my score from 3 to 4 and recommand a weak accept.

**Key Questions For Authors:**

1. Does the filtering procedure affect the privacy accounting or the theoretical DP guarantees?

2. Regarding Weakness 2: Could the authors further clarify which specific aspects of the framework are fundamentally new?

3. (See general weaknesses).

**Limitations:**

yes

**Strengths And Weaknesses:**

Strengths:

1. The interaction between DP noise and adaptive optimizers is a significant and timely research question. The insights regarding second-moment bias are noteworthy.

2. The motivation for the proposed method is intuitive and conceptually well-grounded.

3. The empirical results demonstrate strong performance, which aligns with the provided theoretical analysis.

Weaknesses:

1. While the filter-aware correction is well-motivated, the current theoretical support relies heavily on intuition. The contribution would be significantly strengthened by a more formal and rigorous analysis.

2. The core components—gradient filtering and bias correction—are established techniques widely used in other machine learning contexts. The current approach appears to be a practical integration of these existing ideas rather than a fundamental algorithmic innovation.

3. The paper lacks comprehensive ablation studies to isolate the individual contributions of filtering and bias correction (a point also noted by the authors).

Minor Weaknesses:

1. The presentation could be further refined. Certain sections are terminologically dense, which may make the paper less accessible to readers who are not specialists in Differential Privacy.

---

> ### Author Rebuttal · Authors · 2026-03-30
>
> Thank you for acknowledging the significance of our research question and the strength of our empirical results.
> >W1: The current theoretical support relies heavily on intuition...
> >
> >... a more formal and rigorous analysis.
>
> We respectfully disagree. Our submission includes four formal results with complete proofs, each addressing a specific technical gap:
>
> - Prop. 4.3: Proves $\mathrm{Var}(\tilde{w}_t)=A\sigma_w^2$ via Parseval's identity - a theorem, not an intuitive argument.
> - Cor. 4.4 + App. C.2: Derives $A(\omega)=(2-\omega)/(4-3\omega)$ by solving the Lyapunov equation $\Sigma=M\Sigma M^\top+Q$. This is a non-trivial algebraic result specific to the innovation filter and cannot be obtained by inspection.
> - Lemma C.2: Establishes $\mathrm{Var}(\tilde{w}_{t,i})\leq A(\omega)\sigma_w^2$ for all $t\geq0$ with geometric convergence rate using Loewner-order monotonicity and Gelfand's formula. This is a finite-time bound, not just an asymptotic claim.
> - Prop. C.3: Provides a full second-moment decomposition, with the cross-term $2\mathbb{E}[s_t n_t]$ clearly identified, bounded, and empirically validated (see Table 7).
>
> The filter-aware correction is made precise by the following inequality, which is strict whenever $A(\omega)\sigma_w^2>\varepsilon_v$:
> $$|\mathbb{E}[\hat{v}_t]-\mathbb{E}[s_t^2]| - |\mathbb{E}[\bar{v}_t]-
> \mathbb{E}[s_t^2]| = A(\omega)\sigma_w^2 - \varepsilon_v \geq 0$$
> This guarantees that $\bar{v}_t$ is strictly closer to $\mathbb{E}[s_t^2]$ than $\hat{v}_t$. This is a formal result, not just an intuitive one. Proposition R2 (per-iteration descent, $\mathcal{O}(1/\sqrt{T})$ rate) is included in our response to Reviewer 7zZf(W1) and applies here as well.
> >W2,Q2: The current approach appears to be a practical integration of these existing ideas rather than a fundamental algorithmic innovation.
>
> We clarify that the novelty of FIBER lies not in its individual components but in identifying that their naive combination is actively harmful - and deriving the precise corrections required to resolve this. FIBER provides three contributions absent from prior work:
>
> **Second-order innovation-space filtering.** FIBER applies filtering to the residual stream $\nu_t = g_t -  \tilde{g}_{t-1}$ and integrates it, enabling effective drift-tracking. To our knowledge, no previous DP optimizer has implemented filtering in innovation space.
>
> **Decoupling $(\kappa,\gamma)$ from $\omega$.** In DiSK, $\kappa$ jointly controls temporal smoothing and the two-point mixing weight $a(\kappa,\gamma)$, making independent tuning impossible. FIBER separates them by construction: $(\kappa,\gamma)$ govern only the observation, $\omega$ only the residual recursion - the first explicit decoupling of observation geometry from temporal denoising gain in DP optimization.
>
> **Filter-aware second-moment correction.** To our knowledge, this is the first explicit calibration of AdamW's second-moment estimator to post-filter DP noise statistics. Temporal filtering attenuates DP noise before it enters AdamW's accumulator, making DPAdamBC's correction actively harmful under filtering. The fix requires a per-filter Lyapunov derivation, and not obtainable by inspection. No prior DP optimizer has performed this calibration.
> >W3: ... lacks comprehensive ablation studies to isolate the individual contributions of filtering and bias correction.
>
> The paper presents ablation studies that isolate two of the three components:
>
> - Tables 4–5: Isolate the effect of innovation filtering by comparing FIBER and DiSK under equivalent compute and privacy budgets.
> - Figure 4: Isolates filter-aware correction by comparing full FIBER, FIBER-NO-CORR (correction disabled), and FIBER-BC-CORR (unfiltered correction from [R2]).
>
> Previously, the isolation of the two-point mechanism from filtering and correction together was missing. Table R1 (see our response to Reviewer Qfhr(W1)) addresses this through a complete factorial decomposition:
> - 1-pt FIBER vs. DP-AdamW: isolates the combined effect of innovation filter and filter-aware correction, with the two-point mechanism removed.
> -  Full FIBER vs. DiSK: Isolates the same pair, filter and correction, with the two-point mechanism held constant.
>
> Collectively, Tables 4-5, Figure 4 and Table R1 fully isolate all three components.
> > MW1: Certain sections are terminologically dense...
>
> We agree. The paper will be revised overall, in line with your suggestions.
> >Q1: Does the filtering procedure affect the privacy accounting or the theoretical DP guarantees?
>
> No, as stated in Section 3.3 (footnote 1), all post-privatization operations in FIBER (innovation filtering, bias correction, parameter update) are deterministic functions of the already-privatized gradients $g_t$. By the post-processing property of DP [R1], they cannot reduce the privacy level.
> ## References
> ```
> [R1] The Algorithmic Foundations of Differential Privacy
> [R2] DP-AdamBC: Your DP-Adam Is Actually DP-SGD (Unless You Apply Bias Correction). AAAI 2024
> ```

---

> > ### Author Rebuttal · Reviewer_dgg8 · 2026-04-03
> >
> > We thank the author for thier detailed response which better explain and clarify their method and provide more detailed ablations, I will raise my score to 4 later.

---

### Official Review · Reviewer_Qfhr · 2026-03-16

**Soundness:** 3
**Presentation:** 3
**Significance:** 3
**Originality:** 3
**Overall Recommendation:** 4
**Confidence:** 3

**Summary:**

FIBER proposes a DP optimizer that combines innovation-space temporal filtering with a filter-aware correction to AdamW’s second moment. The key idea is that filtering changes the effective DP noise seen by AdamW, so prior corrections are miscalibrated. The paper shows this matters both theoretically and empirically.

**Compliance With Llm Reviewing Policy:**

Affirmed.

**Key Questions For Authors:**

Do authors think there is a way to get rid of the overheads in the algorithm like the two-point gradients?

**Limitations:**

limitations are well discussed.

**Strengths And Weaknesses:**

Strengths:
1. The novelty is clear, the key idea is a filter-aware AdamW correction under filtered DP noise.
2. Good comparison to prior work, especially DiSK, with useful ablations showing the correction helps.
3. Broad experiments across vision and language tasks, with strongest gains in tighter privacy regimes.

Weaknesses:
1. There is no significant weakness of the approach as improvements over DiSK. However, both FIBER uses two-point gradients just like DiSK, the algorithm is roughly 2x slowdown compared with Adam.

---

> ### Author Rebuttal · Authors · 2026-03-30
>
> Thank you for recognizing the clarity of our novelty and the breadth of our empirical evaluation.
> > W1 : ...the algorithm is roughly 2x slowdown compared with Adam.
>
> **(1) Same gradient steps, faster convergence.** We acknowledge that the two-point construction necessitates two backward passes per training step, but this does not mean there is a 2x slowdown compared to DP-AdamW. FIBER achieves the same accuracy as DP-AdamW in fewer total steps, so the extra cost per step is balanced by faster convergence. For example, on CIFAR-10 at $\varepsilon=4$, FIBER outperforms DP-AdamW at every matched wall-clock time during training (Figure 5(a)). On CIFAR-100 at $\varepsilon=1$, FIBER also leads at every matched gradient-evaluation count (Figure 5(b)). As a result, the real cost of the two-point construction is lower than the per-step overhead might suggest.
>
> **(2) 1.79× in practice, not 2×.** More importantly, the wall-clock overhead is a consistent **1.79×** - not 2× - due to GPU parallelization: concretely, FIBER and DiSK require 7.83 ms/step versus 4.18 ms/step for DP-AdamW, achieving 638K imgs/s versus 1,195K imgs/s (Table 12, Appendix F.2). The gap between the theoretical 2× and the observed 1.79× arises from shared memory reads and partial overlap between the two backward passes.
>
> **(3) Two-point provides curvature information.** This cost is shared with DiSK - the current state-of-the-art DP optimizer - as well as with a broader family of optimizers in non-DP settings, including [R1], and is technically necessary. DP-AdamW uses a single evaluation at $\theta_t$ by design: as a direct adaptation of AdamW to the DP setting, and therefore forgoes the curvature information along the displacement direction $d_{t-1} = \theta_t - \theta_{t-1}$ that the two-point construction provides. The 1.79× overhead of FIBER and DiSK over DP-AdamW is not a flaw in their design - it is precisely the cost of accessing that additional curvature signal. The two-point observation forms:
>
> $$u_t(\xi) = \frac{1-\kappa}{\kappa\gamma}\nabla f(\theta_t+\gamma
> d_{t-1};\xi) + \left(1 -
> \frac{1-\kappa}{\kappa\gamma}\right)\nabla f(\theta_t;\xi)$$
>
> This approach evaluates a geometrically distinct lookahead location that cannot be accessed through a single evaluation at $\theta_t$. Therefore, the second backward pass explores a fundamentally different region of the loss landscape at each step and cannot be replaced by post-hoc filtering of previous observations.
>
> **(4) Removing second pass collapses accuracy.** To verify this, we run 1-point FIBER (single-point DP gradient with innovation filter and filter-aware correction). The innovation filter and filter-aware correction remain theoretically valid under single-point gradients. The performance penalty relative to full FIBER is thus attributable solely to the removal of the two-point construction, as shown in Table R1:
>
> **Tab. R1.** Training CNN5 on CIFAR-10 under DP.
>
> | Config | Compute | $\varepsilon{=}0.5$ | $\varepsilon{=}1$ | $\varepsilon{=}2$ | $\varepsilon{=}4$ | $\varepsilon{=}8$ | Avg $\Delta$ |
> |---|---|---|---|---|---|---|---|
> | DP-AdamW | $1.00\times$ | 44.77 | 49.77 | 54.61 | 60.52 | 63.59 | — |
> | 1-pt FIBER | $1.00\times$ | 46.67 | 53.39 | 56.61 | 63.32 | 66.49 | +2.64 |
> | DiSK | $1.79\times$ | 60.29 | 65.22 | 69.28 | 72.75 | 75.07 | +13.87 |
> | Full FIBER | $1.79\times$ | **66.92** | **70.91** | **74.40** | **75.96** | **76.99** | +18.38 |
>
> The observed average accuracy decrease of 15.74 when omitting the second pass-from +18.38 (Full FIBER) to +2.64 (1-pt FIBER)-directly demonstrates that the curvature information obtained from the lookahead evaluation is a contributor to performance.
>
> **(5) Scalability: Practical across all resource constraints.** With respect to scalability, FIBER demonstrates practicality across a wide spectrum of resource constraints. When two-point evaluation is feasible, FIBER achieves 15 to 30 additional accuracy points per overhead unit compared to DP-AdamW (Table 13, Appendix F.3) under equivalent computational budgets. In scenarios where two-point evaluation is only partially feasible, FIBER can be selectively applied to layers most affected by DP noise. If two-point evaluation is not feasible, the one-point FIBER variant incurs no additional overhead and still provides a gain of 2.64 points over DP-AdamW (Table R1).
> > Q1 : Do authors think there is a way to get rid of the overheads in the algorithm like the two-point gradients?
>
>
> As established in **W1**, the second pass accesses curvature information that is structurally inaccessible from a single evaluation at $\theta_t$. Table R1 confirms that removing it collapses average accuracy gains by 15.74 points (from +18.38 to +2.64 over DP-AdamW). The overhead therefore cannot be eliminated without a commensurate and substantial loss in accuracy.
> ## References
> ```
> [R1] Modality-Aware SAM: Sharpness-Aware-Minimization Driven Gradient Modulation for Harmonized Multimodal Learning NeurIPS 2025
> ```

---

> > ### Author Rebuttal · Reviewer_Qfhr · 2026-04-07
> >
> > I appreciate the authors' response, my questions are fully resolved and I will keep my score.

---

### Decision · Program_Chairs · 2026-04-30

**Decision:**

Accept (regular)

**Comment:**

This paper proposes FIBER, a DP optimizer to temporally filter privatized gradients via three techniques. The authors have experimented it on vision and language tasks under strong privacy guarantee. The empirical results are promising. The reviewers initially had some concerns on novelty and completeness of ablation studies, but they are resolved to reviewers' satisfaction. The main weakness is the overhead, roughly 2x slower than DP-SGD, which can be compensated by faster per-iteration convergence. Nevertheless, the algorithm is overall clean and insightful.